# Judgments of American English male talkers who are perceived to sound gay or straight: Which personal attributes are associated with each group of talkers?

Erik C. Tracy[1]*, Elizabeth D. Young[2,3,4], Kelly A. Charlton[1]

**1** Department of Psychology, University of North Carolina at Pembroke, Pembroke, North Carolina, United States of America, **2** School of Behavioral and Brain Sciences, University of Texas at Dallas, Richardson, Texas, United States of America, **3** Callier Center for Communication Disorders, Dallas, Texas, United States of America, **4** Department of Otolaryngology-Head and Neck Surgery, University of Texas Southwestern Medical Center, Dallas, Texas, United States of America

* erik.tracy@uncp.edu

## Abstract

Upon hearing a spoken utterance, listeners associate certain attributes (e.g., emotional) with self-identified gay male talkers and other attributes (e.g., reserved) with self-identified straight male talkers. In the current study, we explored whether listeners associated additional personal attributes with these types of talkers, and whether different contexts (e.g., listeners being informed of the talker's sexual orientation) affected how strongly listeners associated personal attributes with talkers. Twenty-four talkers (twelve who self-identified as gay and twelve who self-identified as straight) from an established corpus were examined. Notably, previous work found that these talkers' self-described sexual orientation (SO) did not always align with listener-perceived SO (i.e., a self-identified gay talker was perceived as straight sounding, and vice versa). Listeners evaluated these talkers for eight attributes (e.g., boring, confident, intelligent, mad, old, outgoing, sad, and stuck-up) in three contexts: talkers' SO not referenced, talkers' SO truthfully referenced (i.e., listeners were informed that a straight talker was straight), and talkers' SO falsely referenced (i.e., listeners were informed that a straight talker was gay). Results suggested that self-identified gay and straight talkers whom listeners perceived as sounding gay were perceived as confident, mad, stuck-up, and outgoing; self-identified gay and straight talkers whom listeners perceived as sounding straight were perceived as sad and old. Furthermore, listeners' judgments did not differ when the talkers' SO was truthfully referenced, falsely referenced, or not referenced for all attributes except sad and stuck-up. The results indicate that perceived SO generally has the greatest effect on listeners' perception of a talker's attributes and, for most attributes examined, this is the case regardless of whether the listeners are informed (truthfully or falsely) of the talkers' self-identified SO.

**Data availability statement:** The data are contained in a public repository. The link is below: https://osf.io/bc2wm/overview.

**Funding:** The author(s) received no specific funding for this work.

**Competing interests:** The authors have declared that no competing interests exist.

## Introduction

Humans are decidedly social; we have numerous social interactions every day, and many of these interactions last for only a few seconds. Yet, even in these brief encounters, we quickly form impressions about others and decide how to react to them [1]. There is a wealth of research that investigates this process; some researchers focused on how viewers form impressions merely by looking at another individual's face [1], while others have focused on how listeners form impressions by hearing a talker's voice [2] Indeed, listeners do believe they can perceive a broad range of a talker's personal attributes, such as their sexual orientation (SO) [3], age [4], emotional state [5], and personality traits [6]. To illustrate this, consider the following quote from *The Golden Girls* [7].

> Sophia Petrillo (Estelle Getty): Do people treat you differently because you're a lesbian?
>
> Blanche Devereux (Rue McClanahan): Well, most people don't know.
>
> Sophia Petrillo: Really? I would have guessed right off.

This exchange exemplifies how one believes that they can perceive another individual's SO immediately, presumably through how they speak. Could Sophia have formed other impressions of Blanche, besides SO, just based on her voice? Might she think that Blanche is outgoing or intelligent? In fact, research has confirmed that listeners can quickly place talkers into a social category (e.g., gay man or lesbian woman) and associate various personal attributes with them [8,9]. What if Sophia already knew that Blanche was a lesbian woman? Would this label affect how she formed impressions of Blanche? For instance, if Sophia already knew Blanche's SO, Sophia may believe that Blanche is even more outgoing and even more intelligent simply by hearing her voice.

In the current study, we explored which personal attributes are associated with American English male talkers who were classified in a previous study [10] to sound either gay or straight. However, the self-identified SO of these talkers did not necessarily match the perception of their SO by listeners (e.g., talkers self-identified as gay, but were perceived as sounding straight, or vice versa). The first goal of the current investigation was largely exploratory; we investigated which attributes listeners associated with talkers who are perceived to sound either gay or straight, as well as whether actual or perceived SO is more important to listeners when associating attributes with talkers. Not knowing which attributes listeners may associate with gay and straight male talkers, we ran a pilot study in which participants freely described said gay and straight talkers [10,11]. The attributes used in the present study were those attributes that were identified most often by the listeners in the pilot study. The second goal was to determine if labeling talkers' SO would change listeners' perceptions of the talkers' attributes [12]. To that end, we manipulated whether listeners were informed of the talkers' self-identified SO in various experimental conditions; we either did not reference the talkers' SO (Not Referenced), or the talkers' SO was either truthfully or falsely reported (Truthfully Referenced and Falsely Referenced, respectively).

It should be noted that there is a great need to investigate how listeners perceive other gender categories, such as women [13] and trans individuals [14], as well as other SOs [15]. However, this exploratory investigation is concerned with how listeners perceive self-identified gay cis males and straight cis males as a starting point to hopefully generate discussion and be used as a springboard for much-needed research into other populations.

## Perception of SO

Many perceptual studies [2,3,16–18] have found that listeners can perceive a talker's SO solely from the sound of their voice (often termed "auditory gaydar" in the literature). The perception of SO for men has been linked to various acoustic cues, such as pitch variability. Listeners, for instance, may rely on pitch variability to distinguish gay talkers from straight talkers [3,19–21]. The acoustic cues used to produce a stereotyped gay voice may not necessarily be the same acoustic cues that listeners employ when perceiving a talker's self-reported SO [22]. Listeners are also able to form relatively accurate SO judgments about talkers' self-identified SO upon being presented with either a relatively long speech stimulus (i.e., 15 seconds) [3] or a relatively short speech stimulus (i.e., a single phone) [10].

## Perception of attributes

As noted above, many personal attributes of an individual, such as their emotion and attitude, can be perceived from auditory cues alone. Here, we will use the term attributes as an overarching term to describe emotions, attitudes, and other characteristics, such as age, that don't necessarily fall into the emotion or attitude categories. Emotions, such as anger and fear, are often intense and produced in response to a significant event [23,24]. As defined by others [24,25], attitudes, such as confidence and selfishness, are relatively long-lasting beliefs and predispositions that are less intense and more controlled compared to emotions.

The pilot study (described more fully in the methods section) found that certain attributes were often associated with gay male talkers (confident, outgoing, stuck-up), other attributes were often associated with straight male talkers (old, sad, mad, and boring), and one attribute was associated with both groups of talkers (intelligent). Thus, a brief overview regarding these personal attributes specifically for male talkers is given below. We classified sad, mad, and boring as emotions [25,26]. The participants used the term "mad" to describe straight male talkers rather than "angry"; however, we presumed that mad and anger were closely linked in our participants' minds and thus we will discuss the research results for anger. Additionally, we further explored the use of the terms "mad" and "anger" in Experiment 1a. Confident, outgoing, stuck-up, and intelligent were classified as attitudes [6]. Lastly, we will briefly discuss, when appropriate, both listeners' accuracy when identifying the attributes, as well as the acoustic cues necessary to perceive the attributes [25].

The social voice space consists of two dimensions [6,12]. The first dimension is valence, which concerns whether an emotion is positive or negative [26]. All emotions studied here (i.e., sadness, mad, and boring) have a negative valence, while one of the attitudes (stuck-up) also has a negative valence. The remaining attitudes (i.e., confident, intelligent, and outgoing) have a positive valence. The second dimension is dominance, and one of our attitudes (e.g., confident) is seen as being highly dominant. In terms of acoustic cues, a higher average pitch corresponds to an increase in valence, while a decrease in average pitch was found to be more dominant [6].

Listeners can perceive a talker's emotion through various acoustic cues [5,26–28]. For example, listeners may rely on a greater variability in fundamental frequency ($f_o$) [29] and a faster rate of articulation [28] to perceive anger. Listeners may rely on less variability in $f_o$ and a relatively slower rate of articulation to perceive sadness [30]. Finally, listeners often confuse talkers to sound boring with talkers who sound depressed [31]; we presume that the auditory cues that indicate depression are likely to indicate talkers sound boring as well. It was found, for instance, that depressed talkers had less variability in $f_o$ and speak softly [32]. Listeners may rely on these acoustic cues to perceive a boring voice.

Many attitudes are signaled acoustically by changes in prosodic contour and speech rate [25]. However, the acoustic or perceptual features of many individual attitudes, including stuck-up and outgoing have not, to our knowledge, been

studied in any detail. With respect to sounding intelligent, it was discovered that gay men who produced both/s/-fronting and the variable –*ing* (e.g., walking), as opposed to –*in* (e.g., walkin'), were more likely to be labeled as effeminate and educated [16]. Here, we presume that the labels educated and intelligent are closely linked in listeners' minds. Finally, utterances perceived as confident have a relatively large $f_o$ range and a steeper amplitude slope (i.e., the utterance started quiet and eventually got louder) [33,34].

Finally, talker age is perceived through various acoustic cues. For example, listeners reported that older talkers (i.e., over the age of 65) had a slower rate of articulation compared to younger talkers (i.e., under the age of 35) [4]; thus, listeners may rely on rate of articulation to perceive a male talker's relative age [35].

## SO and attributes

Before further consideration of the link between which attributes are associated with gay and straight male talkers, it would be helpful to briefly consider which attributes are associated with gay and straight men in general. To start, there is a stereotype that gay men have attributes associated with women, and lesbian women have attributes associated with men [36]; this stereotype appears to be consistent across different periods of time [37]. Indeed, a group stereotype may affect one's initial impression of another individual; for instance, women are believed to warmer than men, so a gay man might be perceived to be warmer than a straight man even if the gay man displays no indication of warmth [38]. As for specific attributes, gay men are perceived as talkative, gentle, fashionable, artistic and sensitive [11], whereas straight men are perceived as angry [39]. Upon reading a short description of a man, participants indicated that younger men were gay and older men were straight; it was proposed that young men are automatically processed as gay and that elderly men are automatically processed as heterosexual [40]. Gay men, as a subset of men, are viewed as being less competent than men in general [41]. However, gay men are not necessarily a homogenous category; different groups of gay men might have different attributes associated with them. For instance, a leather and biker gay man might be perceived as "macho" and likely to have tattoos, while a flamboyant gay man might be perceived as being loud and dramatic [42].

To our knowledge, there are fewer experiments that have investigated which attributes are associated with gay and straight male talkers, and talkers who are perceived to sound gay and straight (even if their self-identified SO does not match their perceived SO). Aligning with previously discussed research, it was found that listeners assigned gender typical stereotypes to straight male voices and assigned gender atypical stereotypes to gay male voices. For example, straight male talkers are perceived as aggressive, while gay male talkers are perceived as emotional [2]. With respect to the attributes in the current investigation, a stereotypical gay male voice was judged to be more intelligent compared to a stereotypical straight male voice [16], and a gay-sounding man was perceived to be as social, or presumably outgoing, as a straight-sounding man [17]. In a different study [8], gay-sounding and straight-sounding men were rated equally competent (where "competence" was described using three terms: competent, intelligent and skilled). Additionally, gay-sounding men were rated to be more social than heterosexual-sounding men (where "social" was described using the terms: likable, warm, and friendly). While the term *outgoing* was not specifically used here, it seems logical that these listeners might rate gay-sounding men as being more outgoing. To our knowledge, there are no studies that directly investigated the attributes sad, mad, boring, stuck-up, and confident.

Understanding the influence of SO on attributes is of critical importance, particularly if there is an influence of even the impression of a certain SO [39]. In fact, there are many investigations that have found that LGBTQ+ individuals are at higher risk for discrimination compared to non-LGBTQ+ individuals [1,2,8,17,37,38,43–45]. For example, gay and straight men may face discrimination in the workplace [38]; straight men may be turned down for jobs that are typically associated with women, such as nurses, while gay men may not get a job typically associated with men, such as an engineer. Upon seeing photographs of gay and straight men, participants, including experienced hiring managers, were more likely to assign gay men to more caretaking jobs, such as pediatricians and English teachers, and assign straight men to more occupational roles, such as managers and surgeons [43]. Men with stereotypical gay voices may also experience negative

stereotyping [45]. For example, in a study of music teachers, a stereotypical straight voice was rated as more mature, a better leader, and a stronger classroom manager; a stereotypical gay voice was rated as better organized and having higher standards. Finally, gay-sounding job applicants are perceived to be less competent and, thus, not considered for leadership jobs [17]. In sum, merely sounding gay might create a negative impression, and these negative impressions could lead to discrimination; thus, a deeper understanding is needed between which attributes are associated with gay and heterosexual sounding men.

### Knowledge of an individual's self-identified SO

There is one largely unanswered question within the auditory gaydar literature that requires further examination: does one's knowledge of a talker's self-identified SO influence which attributes are associated with said talker? In four different studies [2,8,17,45], listeners judged gay-sounding and straight-sounding male talkers on a variety of dimensions ranging from the effectiveness of the talkers' teaching abilities [45] to the suitability of talkers for a job [17]. For each of these four studies, SO was not explicitly revealed; listeners were not told whether the talker was gay or straight. Even other related studies [43] did not reference an individual's SO when listeners were asked to make a hiring decision. It seems reasonable to assume that this additional piece of knowledge may affect participants' responses when it comes to teaching effectiveness or job suitability.

Studies in related areas provide some insight. For example, knowledge of a speaker's age affected how listeners perceived a vowel [46]. Additionally, if listeners believed that a talker was from Detroit, USA, then they classified vowels as produced by Detroit talkers; if listeners believed that a talker was from Canada, then they classified these same vowels as being produced by a Canadian talker [47]. In both studies, the label itself affected a listener's perception of the utterance. Since an individual's SO might not be readily apparent to others, they might not use this information when forming their initial impressions [1]. However, once an SO label is presented, individuals may use this information to form their impressions.

Recently, researchers have begun to investigate both *social vision* [48] and *social hearing* [2]. It has been theorized that there are two pathways in which social hearing affects listeners' impressions and behaviors toward speakers: a category-based process and a feature-based process. For the category-based process, the speech signal may lead to a social categorization, which in turn activates stereotypes and corresponding behaviors. For the feature-based process, the speech signal may activate stereotypes and corresponding behavior directly, essentially bypassing the categorization [2]. In the present study, the category-based process would predict that a listener would hear a voice and that voice would be perceived as gay; as a result, gay stereotypes (e.g., warmth) are activated in the listener's mind. For the feature-based process, certain stereotypes (e.g., warmth) might be activated simply because the listener perceives the voice as feminine [2].

While the current study does not test this model directly, it may offer some insight. If the category-based process is dominant, then a talker being labeled as either gay or straight would, presumably, result in stronger attributes for that particular SO compared to a talker not being labeled at all. For instance, knowing that a talker is straight might result in stronger mad ratings compared to not knowing that the talker is straight. However, if the feature-based process is dominant, then a talker's SO label, or lack of SO label, should result in equally strong stereotypes. For instance, mad ratings should be relatively equal regardless of the listener's knowledge, or lack of knowledge, of the talker's SO.

### Experimental objectives

The first objective of Experiment 1 was largely exploratory. It was to ascertain which attributes listeners associated with talkers who are perceived to sound either gay or straight, and whether perceived or actual SO most strongly affected attribute perception. The pilot study described in the methods section was used to select several personal attributes for examination for this objective. It has previously been found that a talker's self-identified SO did not always match how

listeners perceived the talker's SO [10], thus, the relationship between perceived and actual SO and attribute perception is of interest. As noted throughout the literature review above, there is limited research examining the various attributes associated with gay and straight talkers; however, as discussed, it is important to understand the potential differences between the perception of these groups of talkers as these may be drivers of discrimination.

The second objective of Experiment 1 was to assess whether a listener's knowledge of a talker's SO would influence which attributes were associated with them. To that end, we manipulated whether listeners were informed of the talkers' self-identified SO; we did not reference the talkers' SO (Not Referenced), or the talkers' SO was either truthfully or falsely reported (Truthfully Referenced and Falsely Referenced, respectively). As indicated previously, if the category-based process is dominant, then an SO label would result in stronger ratings of an attribute compared to no SO label; if the feature-based process is dominant, then attribute ratings should be similar regardless of the SO label, or lack of a label.

Finally, Experiment 1a was included to explore listeners' interpretation of the attribute mad. Given that this attribute has two possible interpretations (angry or crazy), the aim of this experiment was to determine whether the results of Experiment 1 would be largely replicated if the word "mad' was replaced with the word "angry".

## Experiment 1

### Materials and methods

This research was approved by the Institutional Review Board at the University of North Carolina at Pembroke via an expedited review, protocol number 14-08-004. Participant recruitment started on August 4, 2016 and ended on June 8, 2019. Participants were provided an informed consent form by the first author or research assistants. If the participant consented to participate in the study, they signed the consent form with either the first author or a research assistant as a witness.

**Participants.** Two hundred and ten participants who were taking an introductory psychology course at a rural southeastern university participated in the study in exchange for course credit; participants indicated their consent by signing their name to an informed consent form. In a post-experiment questionnaire, we collected demographic information about the participants, such as their age and gender, but regrettably, we did not collect information related to participants' race or ethnicity. However, the student population at said rural southeastern university is racially and ethnically diverse. For example, the student population was approximately 34% White, 30% Black or African American, 12% American Indian or Alaska Native, 10% Hispanic or Latino, 6% Two or More Races, and 2% Asian [49]. We presume that the relatively large participant sample was similar, in this respect, to the larger campus population. The participants' mean age was 19.6 (Range=[18–53]); approximately 95% of participants were between 18 and 23 years old. All participants self-reported that American English was the first language that they learned and used, none reported a history of hearing or speech disorders [50]. These are common inclusion criteria for listeners in these types of perceptual experiments [51].

**Stimuli.** As detailed previously, utterances were provided by 24 male talkers (12 self-identified as straight and 12 self-identified as gay) from the Ohio State University campus and the surrounding Columbus, Ohio metropolitan area [10]. It should be noted that the talkers' self-identified SO did not always align with how listeners perceived them based on results gathered in a previous study [10]. For instance, a self-identified straight talker was sometimes perceived as gay and vice versa. While talkers were required to be from Ohio, the researchers did not control for specific areas of the state and their corresponding regional dialects. The first author, who is a trained phonetician, did not perceive any noticeable differences among the talkers' dialects; we presumed that listeners would not perceive any differences either. Yet, we cannot rule out the possibility that disparities between the listeners' regional dialects and the talkers' regional dialects influenced some listeners' responses [52].

Based on the results of earlier research [10], we selected utterances produced by the six self-identified gay talkers whom listeners were most confident in labeling as gay and the six self-identified gay talkers whom listeners were most

confident in labeling as straight [45]. A similar procedure for selecting utterances was used for the self-identified straight talkers. Here, confidence in a talker's SO was operationalized by using a 7-point scale, where a '7' rating indicated that the listener was very confident that the talker was gay and a '1' rating indicated that the listener was very confident that the talker was straight. Therefore, for instance, we selected the six self-identified gay talkers that had the highest rating on the 7-point rating scale and the six self-identified gay talkers who had the lowest rating. This procedure resulted in utterances from four groups of six talkers: gay talkers who were perceived as gay-sounding, gay talkers who were perceived as straight sounding, straight talkers who were perceived as gay-sounding, and straight talkers who were perceived as straight-sounding.

The utterances produced by these four groups of talkers were concatenated strings of three monosyllabic CVC words that contained two instances of/s/, in line with previous research that listeners relied on/s/ to distinguish between gay and straight male talkers of American English [53]. The two utterances were *dose-wet-soap* and *sell-tone-niece* [54]. Thus, we created 48 different stimuli. In line with others, we used read speech to minimize the possibility that listeners would use the content of spontaneous speech to form their attribute judgments [14].

**Pilot study to identify attributes.**  The purpose of the pilot study was to explore the most common attributes associated with gay and straight talkers. To that end, we selected the seven self-identified gay talkers whom listeners were most confident in labeling as gay and the seven self-identified straight talkers whom listeners were most confident in labeling as straight [10,45]. The utterances produced by these talkers were concatenated strings of three monosyllabic CVC words (e.g., *sad-vein-niece* and *dose-wet-soap*) [54]. Two utterances were created for each talker and there were 28 trials. Upon hearing the utterances, 21 participants, from a rural southeastern university, freely indicated anything that they wanted to about the talkers [11]. Talkers' self-identified SO was not discussed during the study procedures. Based on the results, the most cited attributes were selected for examination in the current study. For instance, sad was indicated 137 times across both SOs, and mad was indicated 56 times across both SOs. While some of these attributes were listed more frequently for gay talkers compared to straight talkers, care was taken to choose attributes that were listed with relatively high frequency across both talker groups. Based on the pilot data, the selected attributes were sad, boring, confident, outgoing, mad, old, intelligent, and stuck-up. We used the attribute mad, instead of angry, because this was the term indicated by participants [27]. Even though there are multiple interpretations of mad (e.g., angry or crazed), we presumed that listeners would associate mad with angry; to confirm this, we examined whether listeners associated mad with angry in Experiment 1a.

**Design.**  To test all the stimuli under the experimental conditions, we created four experimental lists for each of the three groups of listeners (e.g., Not Referenced, Truthfully Referenced, and Falsely Referenced); the three groups of listeners were all presented with the same four lists. Moreover, four attributes (e.g., mad, intelligent, stuck-up, and boring) were assigned to the first two lists, while the other four attributes (e.g., sad, outgoing, old, and confident) were assigned to the remaining two lists. Within a single list, there were 192 trials. Each list was divided into four blocks, and each block corresponded to one of the four attributes (e.g., boring, intelligent) that the listeners would be evaluating; utterances were randomized within each block. However, if a listener evaluated an utterance to be boring and, in a subsequent block, they were tasked to evaluate the same utterances as being intelligent, their earlier boring judgments might influence their intelligent judgments. To account for these potential carryover effects, we counterbalanced the order of assessed attributes. For example, within one list, one group of listeners evaluated utterances on the boring attribute in Block 1 and evaluated utterances on the intelligent attribute in Block 4. In a second list, a second set of listeners evaluated utterances on the intelligent attribute in Block 1 and evaluated utterances on the boring attribute in Block 4. Approximately 15–18 listeners were assigned to each of the twelve lists.

**Procedure.**  Participants were brought into a sound-treated room and seated in front of a computer monitor and mouse. They were instructed to listen carefully to the stimuli over a set of headphones and, after hearing each utterance, to indicate via the mouse how strongly the talker was associated with a particular attribute. There was a column of seven

squares on the left side of the computer screen, numbered from '7' to '1' starting at the top going down. The '7' square was labeled *Strongly associated with this trait* and the '1' square was labeled *Weakly associated with this trait*; the rest of the squares were not labeled other than their corresponding number. Here, in the experiment, we used the term *trait* because, presumably, participants would be more familiar with this term rather than *attribute*. The attribute that the participant judged was written at the top of the computer screen; next to this attribute, for both the Truthfully Referenced and Falsely Referenced conditions, the talkers' SO was displayed. Participants were encouraged to use the entire 7-point scale and to respond as quickly as possible; only participant ratings were collected. After several practice trials, participants were allowed to ask the researcher questions and, at the conclusion of the experiment, participants completed a demographic questionnaire and were debriefed.

**Statistical analyses.** Likert data, such as that used in the current study, are frequently treated as continuous, normally distributed data for the purposes of statistical analyses. However, a seven-point equal-appearing-interval scale should more appropriately be treated as count data, as it increases only in discrete units and is bounded by zero. Moreover, as discussed previously, the talkers were purposefully selected in a way that allowed us to treat perceived SO as a binary; perception of SO is not a binary, but we chose to treat it as such because we were examining the talkers at the extreme ends of the spectrum. Thus, the data were analyzed with a series of mixed-effects Poisson regression models, using a separate model for each attribute as the dependent variable.

Both research aims, or objectives, were answered using separate statistical tests. The first aim, to investigate how strongly participants associated eight attributes (i.e., boring, confident, intelligent, mad, old, outgoing, sad, and stuck-up) with utterances produced by self-identified gay and straight male talkers who sounded both gay and straight, were analyzed using a mixed-effects Poisson regression. To further probe this question, we investigated whether a talker's self-identified SO, or their perceived SO, was the most important factor when listeners formed their judgments about the talkers. The second aim (i.e., whether there were differences in the listeners' attribute ratings depending on whether truthfully or falsely informed of the talker's SO) was likewise examined in two ways. First, it was examined using the fixed effect of condition in the mixed-effects Poisson regression models described above. It was also examined using a second model for each attribute containing two interaction effects, using the same referent categories described above; the first interaction was between condition and perceived SO, and the second interaction was between condition and actual, or self-identified, SO.

Random structures for all mixed-effects models were determined separately, where each model included all considered random effects so long as each a) improved model fit as measured by a −2 log-likelihood test compared to the previous best-fitting model, and b) the data continued to support the more complex model, as evidenced by visual examination of variance components and model estimates. Random effects considered for each model included the following, in order of consideration: a) random intercept of listener; b) random by-listener slope of listener gender (so long as a random intercept of listener improved model fit); c) random intercept of stimulus; d) random intercept of list.

All statistical analyses were completed in R (version 4.3.3). Mixed-effect Poisson regression models were completed using the **glmmTMB** package (version 1.1.9) [55]. Significant interactions were decomposed using the **emmeans** package (version 1.10.0) [56]. We determined that a statistical test reached significance if $p < 0.05$. Finally, it should be noted that given the large number of statistical analyses that were performed, for the most part, we will only be reporting and discussing those analyses that reached statistical significance. All the data sets and R files, containing statistical analyses, are publicly available at the following link: https://osf.io/bc2wm/overview Those readers who wish to discuss all statistical analyses contained in the R files, such as those that did not reach statistical significance, should contact the first author.

## Results

**Aim 1. Strength of Association of attributes with actual and perceived sexual orientation.** As a reminder, the first aim was to determine how strongly participants associated eight attributes with utterances produced by self-identified gay

and straight male talkers who sounded both gay and straight. Descriptive statistics of each attribute by talker actual SO, perceived SO, and condition are displayed in Table 1. An examination of this table reveals relatively small differences in ratings between the utterances produced by gay and straight talkers (i.e., actual SO) for all attributes. This was confirmed in the results for the fixed effects of actual SO as examined in the mixed-effects Poisson regression, which are summarized in Table 2. As can be seen in Table 2 and Fig 1, none of the fixed effects of actual SO reached statistical significance. In contrast, Table 1 reveals several attributes with substantial differences in ratings between utterances produced by talkers who were perceived as sounding straight and talkers who were perceived as sounding gay, regardless of the talker's actual SO, which can be seen in Fig 2. For example, the largest magnitude difference was for stuck-up, where talkers perceived as sounding straight were given an average rating of 3.21 and talkers perceived as gay were given an average rating of 4.44

**Table 1. Descriptive statistics for each attribute.**

| Sad | | Mean | SD | Mad | | Mean | SD |
|---|---|---|---|---|---|---|---|
| Actual SO | Gay | 3.69 | 2.02 | Actual SO | Gay | 3.47 | 1.89 |
| | Straight | 3.50 | 1.83 | | Straight | 3.48 | 1.92 |
| Perceived SO | Gay | 3.18 | 1.79 | Perceived SO | Gay | 3.86 | 1.93 |
| | Straight | 4.01 | 1.97 | | Straight | 3.08 | 1.80 |
| Experiment | Not Referenced | 3.30 | 1.98 | Experiment | Not Referenced | 3.48 | 1.91 |
| | Truthfully Referenced | 3.74 | 1.89 | | Truthfully Referenced | 3.41 | 1.87 |
| | Falsely Referenced | 3.71 | 1.89 | | Falsely Referenced | 3.54 | 1.94 |
| **Confident** | | | | **Intelligent** | | | |
| Actual SO | Gay | 4.16 | 1.96 | Actual SO | Gay | 3.99 | 1.74 |
| | Straight | 4.29 | 1.81 | | Straight | 4.3 | 1.70 |
| Perceived SO | Gay | 4.67 | 1.77 | Perceived SO | Gay | 4.06 | 1.69 |
| | Straight | 3.78 | 1.89 | | Straight | 4.23 | 1.77 |
| Experiment | Not Referenced | 4.16 | 1.9 | Experiment | Not Referenced | 4.02 | 1.75 |
| | Truthfully Referenced | 4.18 | 1.89 | | Truthfully Referenced | 4.21 | 1.67 |
| | Falsely Referenced | 4.32 | 1.86 | | Falsely Referenced | 4.19 | 1.77 |
| **Outgoing** | | | | **Stuck-up** | | | |
| Actual SO | Gay | 3.88 | 1.95 | Actual SO | Gay | 4.07 | 2.08 |
| | Straight | 3.82 | 1.77 | | Straight | 3.59 | 1.95 |
| Perceived SO | Gay | 4.43 | 1.81 | Perceived SO | Gay | 4.44 | 1.96 |
| | Straight | 3.28 | 1.74 | | Straight | 3.21 | 1.90 |
| Experiment | Not Referenced | 3.85 | 1.88 | Experiment | Not Referenced | 3.71 | 2.10 |
| | Truthfully Referenced | 3.85 | 1.86 | | Truthfully Referenced | 3.87 | 1.93 |
| | Falsely Referenced | 3.85 | 1.86 | | Falsely Referenced | 3.88 | 2.07 |
| **Old** | | | | **Boring** | | | |
| Actual SO | Gay | 3.44 | 1.62 | Actual SO | Gay | 4.20 | 2.02 |
| | Straight | 3.80 | 1.71 | | Straight | 3.99 | 1.82 |
| Perceived SO | Gay | 3.09 | 1.56 | Perceived SO | Gay | 3.74 | 1.85 |
| | Straight | 4.14 | 1.61 | | Straight | 4.45 | 1.93 |
| Experiment | Not Referenced | 3.48 | 1.69 | Experiment | Not Referenced | 4.05 | 1.99 |
| | Truthfully Referenced | 3.69 | 1.65 | | Truthfully Referenced | 4.11 | 1.81 |
| | Falsely Referenced | 3.67 | 1.67 | | Falsely Referenced | 4.12 | 1.99 |

Notes: SO = sexual orientation. SD = standard deviation.

**Table 2. Summary of mixed-effects Poisson regression fixed effects models for each attribute.**

| Attribute | Fixed effect | Estimate | RR | RR 95% CI | z | p |
|---|---|---|---|---|---|---|
| Sad | | | | | | |
| | Actual SO[1] | −0.03 | 0.97 | [0.78, 1.22] | −0.23 | .822 |
| | Perceived SO[2] | −0.23 | 0.79 | [0.63, 0.99] | −2.02 | **.044** |
| | Truthfully Referenced[3] | 0.13 | 1.14 | [1.04, 1.24] | 2.98 | **.003** |
| | Falsely Referenced[3] | 0.12 | 1.13 | [1.04, 1.23] | 2.95 | **.003** |
| Confident | | | | | | |
| | Actual SO[1] | 0.07 | 1.07 | [0.87, 1.32] | 0.61 | .540 |
| | Perceived SO[2] | 0.24 | 1.27 | [1.03, 1.56] | 2.22 | **.026** |
| | Truthfully Referenced[3] | 0.00 | 1.00 | [0.95, 1.06] | 0.16 | .874 |
| | Falsely Referenced[3] | 0.04 | 1.04 | [0.99, 1.09] | 1.54 | .124 |
| Outgoing | | | | | | |
| | Actual SO[1] | 0.01 | 1.01 | [0.81, 1.27] | 0.11 | .910 |
| | Perceived SO[2] | 0.31 | 1.37 | [1.09, 1.71] | 2.73 | **<.001** |
| | Truthfully Referenced[3] | 0.00 | 1.00 | [0.94, 1.06] | 0.01 | .992 |
| | Falsely Referenced[3] | 0.00 | 1.00 | [0.94, 1.06] | 0.00 | .998 |
| Old | | | | | | |
| | Actual SO[1] | 0.10 | 1.10 | [0.95, 1.27] | 1.32 | .186 |
| | Perceived SO[2] | −0.30 | 0.74 | [0.64, 0.85] | −4.11 | **<.001** |
| | Truthfully Referenced[3] | 0.07 | 1.07 | [0.99, 1.16] | 1.66 | .097 |
| | Falsely Referenced[3] | 0.06 | 1.06 | [0.98, 1.15] | 1.49 | .137 |
| Mad | | | | | | |
| | Actual SO[1] | 0.00 | 1.00 | [0.87, 1.14] | −0.05 | .962 |
| | Perceived SO[2] | 0.22 | 1.25 | [1.09, 1.43] | 3.13 | **.002** |
| | Truthfully Referenced[3] | −0.01 | 0.99 | [0.89, 1.1] | −0.20 | .840 |
| | Falsely Referenced[3] | 0.02 | 1.02 | [0.92, 1.14] | 0.44 | .661 |
| Intelligent | | | | | | |
| | Actual SO[1] | 0.09 | 1.09 | [0.94, 1.27] | 1.16 | .247 |
| | Perceived SO[2] | −0.03 | 0.97 | [0.84, 1.13] | −0.33 | .741 |
| | Truthfully Referenced[3] | 0.05 | 1.05 | [1.0, 1.1] | 1.84 | .066 |
| | Falsely Referenced[3] | 0.04 | 1.04 | [0.99, 1.1] | 1.58 | .114 |
| Stuck-Up | | | | | | |
| | Actual SO[1] | −0.12 | 0.89 | [0.79, 1.01] | −1.84 | .066 |
| | Perceived SO[2] | 0.32 | 1.38 | [1.22, 1.56] | 5.12 | **<.001** |
| | Truthfully Referenced[3] | 0.05 | 1.05 | [0.97, 1.14] | 1.20 | .232 |
| | Falsely Referenced[3] | 0.05 | 1.05 | [0.97, 1.15] | 1.19 | .234 |
| Boring | | | | | | |
| | Actual SO[1] | −0.03 | 0.97 | [0.81, 1.15] | −0.39 | .695 |
| | Perceived SO[2] | −0.17 | 0.85 | [0.71, 1.0] | −1.90 | .057 |
| | Truthfully Referenced[3] | 0.02 | 1.02 | [0.97, 1.06] | 0.69 | .490 |
| | Falsely Referenced[3] | 0.02 | 1.02 | [0.97, 1.07] | 0.77 | .440 |

Notes: SO = Sexual Orientation. RR = Rate Ratio. CI = Confidence Interval. [1]Reference category = gay. [2]Reference category = perceived as sounding straight. [3]Reference category = Not Referenced condition. Bolded p values indicate the value was < .05.

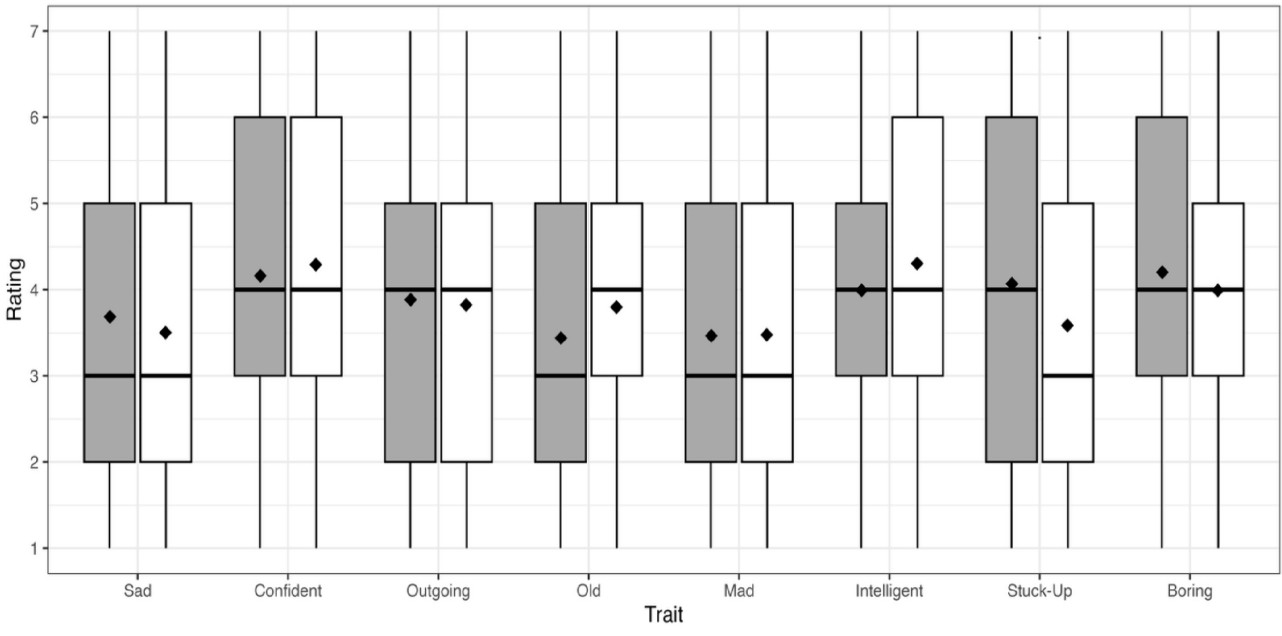

**Fig 1. Average listener ratings of each attribute by talker self-identified SO.** Listeners rated each attribute using a 7-point Likert scale, where 7 represented "Strongly associated with this trait" and 1 represented "Weakly associated with this trait". Solid lines within the boxplots represent median values, while diamonds represent mean values.

(difference = 1.23 Likert scale points). This was again confirmed with the regression models; as seen in Table 2, there was a significant effect of perceived SO for the following attributes: sad, confident, outgoing, old, mad, and stuck-up. The effect was positive (i.e., the attribute was more likely to be perceived for talkers who were perceived as gay) for confident, outgoing, mad, and stuck-up. The effect was negative (i.e., the attribute was less likely to be perceived for talkers who were perceived as gay) for sad and old. As for the two remaining attributes, boring approached significance and the effect was negative, whereas intelligent was not significant. Furthermore, consistent with the descriptive statistics, the magnitude of the effect was largest for stuck-up, where the rate ratio (RR) predicts an increase of 1.38 Likert scale points in this attribute as perceived SO changed from straight to gay. This is followed closely by outgoing, where the RR predicts an increase of 1.37 Likert scale points as perceived SO changed from straight to gay. Thus, the results of Aim 1 indicated that none of the attributes were significantly associated with a talker's actual SO; certain attributes (i.e., confident, outgoing, mad, and stuck-up) were significantly associated with utterances produced by talkers perceived to sound gay, and other attributes (i.e., sad and old) were significantly associated with utterances produced by talkers perceived to sound straight.

**Aim 2. Differences in attribute ratings by listening condition.** The second aim was to determine whether there were differences in the listeners' attribute ratings depending on if they were truthfully or falsely informed of the talker's SO. This was accomplished, first, by examining the fixed effect of condition in the mixed-effects Poisson regression models, which is described in Aim 1 and displayed in Table 2. Moreover, we used an additional model that contained two interaction effects; the first interaction was between condition and perceived SO, and the second interaction was between condition and actual SO.

With respect to the model described in Table 2, the fixed effect of condition was significant for the sad attribute only. This effect for sad was significant for both the Truthfully Referenced and Falsely Referenced conditions, and was positive

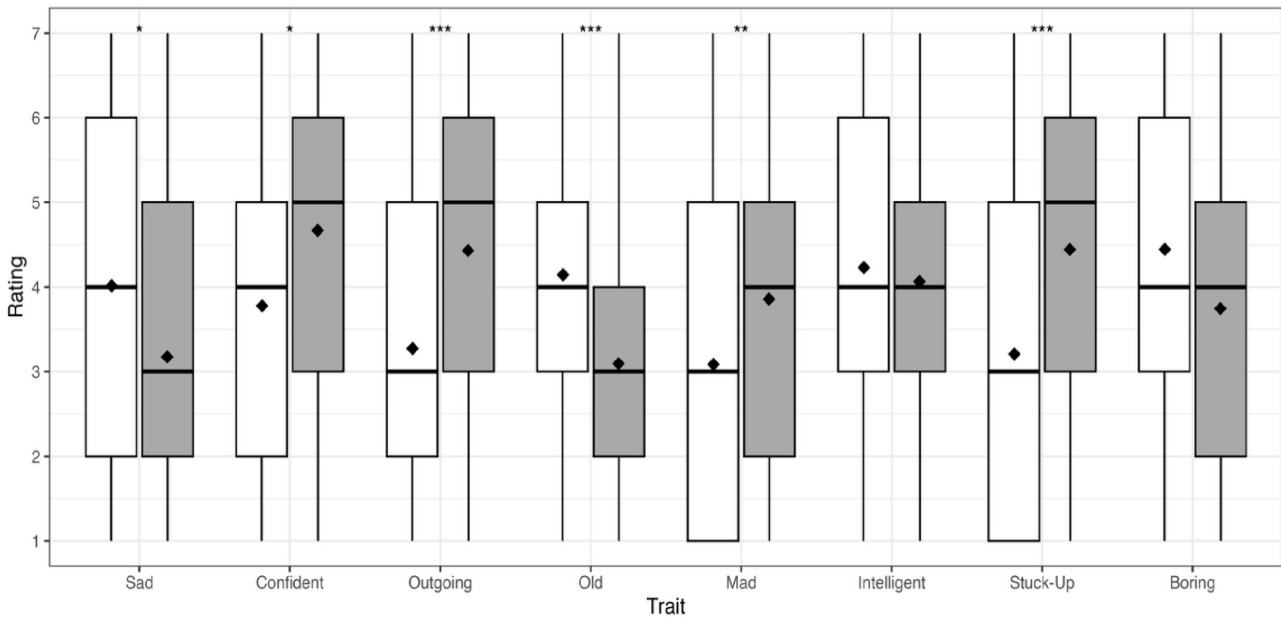

**Fig 2. Average listener ratings of each personal attribute by talker perceived SO.** Listeners rated each attribute using a 7-point Likert scale, where 7 represented "Strongly associated with this trait" and 1 represented "Weakly associated with this trait". Solid lines within the boxplots represent median values, while diamonds represent mean values. * = significant at $p < .05$, ** = significant at $p < .01$, *** = significant at $p < .001$.

in both cases, indicating that, compared to the Not Referenced condition, listeners were more likely to associate the attribute sad with utterances produced by talkers when listeners were both truthfully and falsely told of the talkers' SO. None of the other seven attributes' fixed effects of condition were significant. With respect to all the attributes except sad, this result suggests that the speech signal itself was driving participants' attribute perceptions rather than their knowledge of the talkers' SO.

The interaction between actual SO and condition was significant for confident in the Truthfully Referenced condition ($\beta = -0.07$, RR = 0.93, $z(5031) = -2.02$, $p = .044$), stuck-up in the Falsely Referenced condition ($\beta = -0.13$, RR = 0.88, $z(5031) = -3.57$, $p < .001$), and boring for both conditions ($\beta > 0.06$, RR > 1.07, $z(5031) > 2.03$, $p < .042$). However, after correcting for multiple comparisons using the Sidak method, none of the follow-up contrasts for these interactions reached statistical significance, indicating that the magnitude of these interaction effects was likely small [57].

Finally, the interaction between perceived SO and condition was significant for sad for both conditions ($\beta > 0.09$, RR > 1.04, $z(5031) > 2.43$, $p < .015$) and stuck-up in the in the Falsely Referenced Condition ($\beta = 0.09$, RR = 1.10, $z(5031) = 2.47$, $p = .014$). Significant follow-up contrasts are shown in Table 3, with row numbers labeled and referenced throughout the following two paragraphs. To briefly summarize these results, there was a significant decrease in ratings of sad for listeners who heard utterances produced by talkers perceived as gay in the Not Referenced condition compared to all other talkers and all other conditions (e.g., Rows 6–9). These results suggest that, in the absence of any information regarding SO, listeners perceived utterances produced by talkers who sounded gay to be less sad compared to utterances from all other talker groups when information regarding SO was provided; these results align with the findings from the model described in Table 2. Further, these findings indicate that listeners rated utterances produced by talkers perceived as sounding gay as significantly more sad when they had information regarding their SO, regardless of whether that information was true or false. This interaction effect, as well as the significant contrasts, are displayed in Fig 3.

**Table 3. Follow-up pairwise contrasts for significant perceived SO x Condition interaction effects; also illustrated in Figs 3 and 4.**

| | Sad | | | Stuck-Up | | |
|---|---|---|---|---|---|---|
| | Difference | z | p | Difference | z | p |
| 1. Straight – Not Referenced// Gay – Not Referenced | 1.36 | 2.59 | .099 | 0.74 | −4.22 | **<.001** |
| 2. Straight – Not Referenced// Straight – Truthfully Referenced | 0.93 | −1.70 | .531 | 0.94 | −1.34 | .762 |
| 3. Straight – Not Referenced// Gay – Truthfully Referenced | 1.12 | 0.93 | .938 | 0.71 | −4.24 | **<.001** |
| 4. Straight – Not Referenced// Straight – Falsely Referenced | 0.92 | −1.87 | .419 | 1.00 | 0.04 | 1.000 |
| 5. Straight – Not Referenced// Gay – Falsely Referenced | 1.14 | 1.06 | .898 | 0.68 | −4.81 | **<.001** |
| 6. Gay – Not Referenced// Straight – Truthfully Referenced | 0.68 | −3.10 | **.024** | 1.27 | 2.95 | **.038** |
| 7. Gay – Not Referenced// Gay – Truthfully Referenced | 0.83 | −3.99 | **.001** | 0.96 | −0.92 | .943 |
| 8. Gay – Not Referenced// Straight – Falsely Referenced | 0.68 | −3.16 | **.020** | 1.35 | 3.72 | **.003** |
| 9. Gay – Not Referenced// Gay – Falsely Referenced | 0.84 | −3.73 | **.003** | 0.91 | −1.94 | .376 |
| 10. Straight – Truthfully Referenced// Gay – Truthfully Referenced | 1.21 | 1.65 | .567 | 0.76 | −3.98 | **.001** |
| 11. Straight – Truthfully Referenced// Straight – Falsely Referenced | 0.99 | −0.14 | 1.000 | 1.07 | 1.43 | .710 |
| 12. Straight – Truthfully Referenced// Gay – Falsely Referenced | 1.23 | 1.70 | .533 | 0.72 | −4.12 | **.001** |
| 13. Gay – Truthfully Referenced// Straight – Falsely Referenced | 0.82 | −1.62 | .583 | 1.41 | 4.32 | **<.001** |
| 14. Gay – Truthfully Referenced// Gay – Falsely Referenced | 1.02 | 0.34 | .999 | 0.95 | −1.11 | .876 |
| 15. Straight – Falsely Referenced// Gay – Falsely Referenced | 1.24 | 1.83 | .444 | 0.68 | −5.56 | **<.001** |

Note: "Gay" and "Straight" throughout the above table refer to perceived SO, regardless of the talker's self-identified SO. Bolded p values indicate the value was <.05.

As illustrated in Fig 4, the pattern of significant contrasts was more complex for stuck-up. The difference between this attribute was significant between talkers who were perceived as sounding straight and gay in all three conditions (Rows 1, 10, and 15), signifying a strong effect of perceived SO on the perception of being stuck-up, regardless of condition. All other contrasts reveal that the perception of the attribute was higher for utterances produced by talkers who were perceived as gay than for utterances produced by talkers who were perceived as straight (Rows 3, 5, 6, 8, 12, and 13). None of the contrasts indicating a change in the information listeners had regarding SO altered their perception of who sounded stuck-up were significant (i.e., the contrasts that compared a single perceived orientation across conditions, such as contrasts 2, 4, 7, 9, or 11). Overall, therefore, these results suggest that, regardless of the listening condition, listeners perceived utterances produced by talkers who were perceived as gay as being more stuck-up compared with talkers who were perceived as straight.

## Experiment 1a

The purpose of this experiment was to determine if participants interpreted mad to mean angry, rather than alternate interpretations such as crazy. We reran a portion of Experiment 1 and substituted the word angry for mad. We hypothesized that if the averages for mad, from Experiment 1, were like the averages for angry, then participants in Experiment 1 interpreted mad as angry.

### Materials and methods

**Participants.** A new sample of twenty-five participants, who participated in the experiment in exchange for course credit, had similar characteristics to the sample of participants from Experiment 1.

**Stimuli.** The two stimuli used in this experiment, *dose-wet-soap* and *sell-tone-niece*, were the same stimuli used in Experiment 1.

**Design.** We created one experimental list for participants in which the talkers' SO was not referenced. We assigned four of the eight attributes (e.g., angry, intelligent, stuck-up, and boring) to this list. Within this list, there were 192 trials.

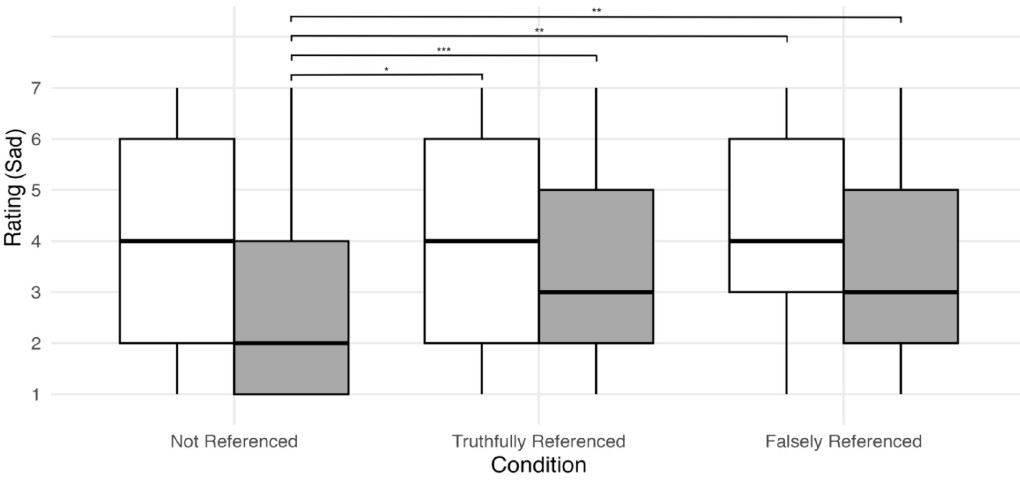

**Fig 3. Interaction effects between perceived SO and perception of the "sad" attribute.** Listeners rated the sad attribute using a 7-point Likert scale, where 7 represented "Strongly associated with this trait" and 1 represented "Weakly associated with this trait". In the three conditions, listeners either received no information about the talkers' SO (Not Referenced), accurate information about the talkers' SO (Truthfully Referenced), or inaccurate information about the talker's SO (Falsely Referenced). * = significant at $p < .05$, ** = significant at $p < .01$, *** = significant at $p < .001$.

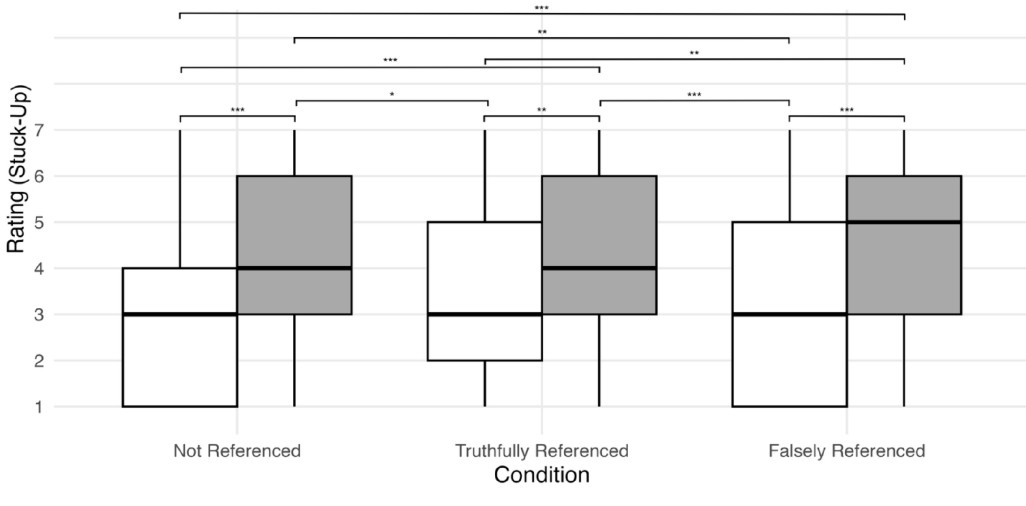

**Fig 4. Interaction effects between perceived SO and perception of the "stuck-up" attribute.** Listeners rated the stuck-up attribute using a 7-point Likert scale, where 7 represented "Strongly associated with this trait" and 1 represented "Weakly associated with this trait". In the three conditions, listeners either received no information about the talkers' SO (Not Referenced), accurate information about the talkers' SO (Truthfully Referenced), or inaccurate information about the talker's SO (Falsely Referenced). * = significant at $p < .05$, ** = significant at $p < .01$, *** = significant at $p < .001$.

The list was divided into four blocks, and these blocks corresponded to which attribute (e.g., angry, intelligent, stuck-up, and boring, respectively) the listeners judged. Within each block, we randomized the utterances.

**Procedure.** The procedure for Experiment 1a was identical to the procedure for Experiment 1.

## Results

We analyzed the data to determine if participants interpreted mad as angry. To accomplish this, the angry ratings in the present experiment were compared against the mad ratings from Experiment 1. The average ratings for angry were 3.59 ($SD = 1.98$) and the average ratings for mad were 3.48 ($SD = 1.91$). The difference between the ratings was not significant, $t(2686) = -1.54$, $p = 0.12$. The average ratings for angry and mad were also compared across actual and perceived SO for gay and straight talkers, as displayed in Table 4 below. None of these contrasts were significant ($t(29) \leq 1.06$, p ≥ 0.296). Based on these results, we concluded that mad was likely to have similar interpretation as angry.

## Discussion

The present experiment had two aims. The first aim, which was largely exploratory, was to determine how strongly listeners associated certain attributes (i.e., boring, confident, intelligent, mad, old, outgoing, sad, and stuck-up) with utterances produced by self-identified gay and straight male talkers who were perceived as sounding gay or straight. Second, we investigated how listeners' perceptions of these attributes might change based on their knowledge of the talkers' SO (i.e., Not Referenced, Truthfully Referenced, or Falsely Referenced).

There are three important results. One, as detailed in Aim 1, some attributes were more associated with one group of talkers compared to the other group of talkers. For example, the attributes confident, outgoing, mad, and stuck-up were significantly more associated with utterances produced by talkers perceived to sound gay, while the attributes sad and old were significantly more associated with utterances produced by talkers perceived to sound straight. Of the two remaining attributes, boring approached significance (i.e., listeners trended towards associating this attribute with utterances produced by talkers perceived as straight) and intelligence was not significant (i.e., listeners did not associate this attribute with utterances produced by either group of talkers).

Two, the talkers' perceived SO, and not their self-identified SO, appeared to be the more important factor for how listeners formed their judgments for all of the studied attributes except boring and intelligent. While boring approached significance (i.e., listeners' ratings trended towards showing differences in this attribute based on perceived SO), intelligent was not significant (i.e., listeners' ratings did not differ based on perceived SO). Stated differently, listeners trended towards perceiving straight-sounding talkers as being more boring compared to gay-sounding talkers; listeners found both gay- and straight-sounding talkers to be equally intelligent. One possible reason for this finding is that listeners were deliberately avoiding high or low ratings for certain attributes, or certain talkers, to avoid appearing biased. For instance, listeners might want to be accepting of different populations, and thus they would not describe a certain group of people as being less intelligent than another group. This possibility is further discussed in the Implications for the Workplace section.

Three, for the attributes confident, outgoing, mad, old, boring, intelligent, and stuck-up, listeners' ratings did not change based on the listening condition (i.e., whether the talker's SO was referenced). In terms of the *social hearing* model [2], there are two pathways in which social hearing can impact listeners' impressions of talkers. The first pathway is a category-based process in which the speech signal leads to social categorization, which in turn activates stereotypes. The second pathway activates stereotypes directly and bypasses the categorization. We predicted that if listeners rely on the

**Table 4. Two-sample t-tests comparing the attributes mad and angry across perceived and actual SO.**

|  |  | Mad | Angry | t (df) | p |
|---|---|---|---|---|---|
|  |  | Mean (SD) | Mean (SD) |  |  |
| Actual SO | Gay | 3.47 (1.89) | 3.49 (2.00) | 0.05 (29) | 0.962 |
|  | Straight | 3.48 (1.92) | 3.70 (1.95) | 0.53 (29) | 0.598 |
| Perceived SO | Gay | 3.86 (1.93) | 3.65 (1.94) | 0.51 (29) | 0.613 |
|  | Straight | 3.08 (1.80) | 3.53 (2.02) | 1.06 (28) | 0.296 |

categorization pathway, then knowledge of the talkers' SO might result in stronger ratings. For example, if listeners have a stereotype that straight men are old, then listeners should indicate, via the rating scale, that they are more confident in their *old* ratings compared to when the listeners had no knowledge of the talkers' SO. For seven of the eight attributes, this was not the finding. Rather, there were no differences in ratings if listeners had knowledge, or no knowledge, of the talkers' SO. In fact, even when we disguised the talkers' SO (i.e., Falsely Referenced Condition), listeners' ratings did not change compared to when listeners had no knowledge of the talkers' SO (i.e., No Referenced Condition). Therefore, our results suggest that, for the most part, the speech signal itself can directly activate stereotypes.

The results for sad, in contrast, did not align with the other attributes. In the absence of any SO labels, listeners rated gay-sounding listeners as being significantly less sad compared to other listening conditions, such as straight-sounding talkers with no SO label and all talkers with a SO label, whether it be true or false. One interpretation of this result is that, in the absence of any information, a gay-sounding voice is not perceived as sounding sad compared to straight-sounding voices. However, there are also several alternate interpretations. First, attributes may work along different pathways in social hearing model [2]. Perhaps when listeners are asked to judge some attributes (e.g., outgoing), the voice itself could activate the stereotypes; however, when asked to judge other attributes (e.g., sad), the voice activates a SO categorization, which in turn, activates the stereotypes. Hence, when SO is made apparent, this SO label activates the sad stereotype. One way to test this hypothesis is to ask participants, in the absence of any vocal cues, whether they associate sadness with gay or straight men. Another possibility is that there is a stimulus-specific effect occurring for this experiment. A concatenated string of three monosyllabic words (i.e., the stimuli used here) presumably does not contain a lot of pitch variability, and less pitch variability is an acoustic feature that cues sadness [30]. However, it should be noted that, within the pilot study, which also used three monosyllabic words as stimuli, listeners freely associated our talkers with the attribute sad. Despite this, it is possible that different speech stimuli would lead to different results. Thus, the most likely explanation given the current results is that, in the absence of any identifying information, listeners perceive utterances produced by gay-sounding talkers as being less sad compared to utterances produced by straight-sounding talkers.

## Implications for the workplace

There are social implications for the results of the current study. As mentioned previously, LGBTQ+ individuals, compared to non-LGBTQ+ individuals, are at a higher risk for discrimination during hiring practices [43] and gay men may also be viewed negatively in the classroom [45]. Indeed, this type of discrimination appears to be quite common in the workplace [58,59]. The evidence suggests that this discrimination appears in both Western and non-Western societies, and, across both these societies, gay men are disliked more than lesbian women [60]. An overarching finding is that gender, sexual, and heterosexist harassment are linked together [58,59]. Sexual harassment is defined as unwanted sex-behavior at work that the intended target perceives as both threatening and offensive; heterosexist harassment is defined as behaviors and/or thoughts that deny, malign, and stigmatize any non-heterosexual behaviors, identities, or relationships [58]. Indeed, the link between gender and heterosexist discrimination should not be surprising given that across many domains (e.g., legal, historical, cultural, psychological), people view lesbian women, gay men, and bisexual individuals (LGB) as failing to conform to traditional gender expectations [44]; gender inequality and a lack of LGB rights align with one another [58,61]. Often, sexual and/or gender harassment can affect an employee's wellbeing in many ways, including increased job dissatisfaction, procrastination, decreased loyalty to employers, and taking time off work [59]. With respect to the current study, current or future employees who are perceived as gay, regardless of their self-identified SO, may experience harassment at work. For example, as part of the interview process, an employer may conduct a phone interview with potential employees, who may be perceived as gay and, therefore, stuck-up. This might not be a personality attribute that is desirable to an employer, and the employee thus may not be hired based solely on how their voice is perceived.

Since SO is not always an obvious identity [1,39], this harassment may extend to individuals who do not self-identify as LGBTQ+. For instance, a self-identified straight man, who is perceived as sounding gay, may be considered less

competent and not considered for a leadership role compared to other straight men who are perceived as sounding straight [17,38]. Providing employers and supervisors additional education regarding how their thoughts and behaviors could be guided by the way that their employees look and sound [1], then it may aid in reducing workplace discrimination.

### Additional considerations and interpretations

We want to be mindful of the conclusions that we draw from our data. One consideration is whether the talkers themselves would self-identify with these attributes. For example, utterances produced by talkers perceived as straight were rated as sad, and utterances produced by talkers perceived as gay were rated as stuck-up – but do these talkers self-identify as being sad or stuck-up? Do they perceive their own speech as sounding sad or stuck-up? Earlier research suggested that labels assigned to talkers might not align with how talkers describe themselves [62]. For example, after rating talkers on a variety of attributes, listeners were often in agreement with one another; if one listener rated a talker as aggressive, then other listeners also rated this same talker as aggressive. However, listeners' ratings did not align with how the talkers described themselves; talkers attributed as aggressive did not describe themselves as aggressive. We should therefore presume that listeners' perceptions of the talkers' speech may not be reflective of how the talkers self-identify.

A second consideration is that many of the attributes used in the current study are what are known as "polar adjectives", which are terms that tend to come in pairs and contain opposite meanings (e.g., sad and happy) [63]. However, just because a single polar adjective was included in the study (e.g., old) does not necessarily mean that talkers would be rated in the opposite direction for the other polar adjective not included in the study (e.g., young). For instance, we found that utterances produced by talkers attributed as straight are perceived as sounding more sad compared with utterances produced by talkers attributed as gay, but this does not necessarily mean that talkers attributed as gay are perceived as sounding more happy. While it may be true that the utterances produced by talkers attributed as gay could be perceived as sounding more happy, the current data does not directly support this assertion. Additional investigations are needed to make further determinations for all our attributes; thus, we urge caution in over-interpreting the results.

### Limitations and future directions

There are methodological limitations to the current study that, if corrected, could lead to further insight. First, we do not wish to overgeneralize our findings. For example, our stimuli were utterances consisting of a concatenated string of three monosyllabic words produced by talkers who were perceived to sound gay or straight; the talkers produced these utterances in a careful manner. Thus, the stimuli themselves do not consider the full range of gay and straight speech; we do not wish to presume that all gay men, for instance, would be perceived as stuck-up or all straight men would be perceived as old. However, even considering these constraints, we did discover relatively consistent impressions of the talkers' identity. Future experiments could further investigate other types of gay and straight speech, either from a wider range of talkers and/or other types of speech (e.g., read paragraphs, spontaneous speech, etc.), and how these additional speech samples could correspond to various attributes.

Moreover, while we presume that our listener population was racially and ethnically diverse [49], the age of our listener population was not diverse. Given the homogeneous age of our relatively young listener sample (i.e., 95% of listeners in Experiment 1 were between 18 and 23 years old), it was not possible to draw any conclusions about how older participants associate attributes with SO. It is possible that older and younger individuals have different interactions with gay and straight individuals (e.g., social interactions and media exposure), and these different encounters could lead to different attributes associated with each group. For example, a younger individual might see gay individuals at college parties, and assume that gay individuals are outgoing, whereas an older individual might only see gay individuals at work and assume that they are not as outgoing. There are also potential generational differences between listeners in their overall perception of the LGBTQ+ community, which have come alongside the general cultural shift towards increased public support and acceptance and the passage of LGBTQ+ affirming legislation in the United States where the study took place

[64]. Alongside this cultural shift, there may be increased internal and external pressure for individuals to be observant of their own behaviors towards the LGBTQ+ community to avoid offense. Thus, particularly in the listening conditions where listeners were informed of the talkers' SO (either truthfully or falsely), listener's own perception of the LGBTQ+ community, and their desires to minimize stigma or bias, may have influenced the results. Future work could investigate this further by repeating the experiment with listeners across multiple age groups who complete questionnaires examining their interactions with and perceptions of the LGBTQ+ community.

Another consideration for recruiting primarily young adult listeners is the differences in how older and younger listeners perceive emotion from voice. Older individuals (60–84 years), compared to younger individuals (17–29 years), were relatively inaccurate at identifying sadness and anger [65]. While one possible reason for this is due to age-related changes in certain parts of the brain, other literature has discussed the presence of a "positivity effect" in older listeners where the intensity of negative emotions is reduced due to their shift of motivation and attention towards positive events rather than negative events [66]. Thus, our participants may have been more attuned to judgments of emotion, particularly negative emotions, simply because they were younger. Additionally, other age-related changes such as hearing loss or changes in cognitive processing could contribute to differences in the perception of many attributes discussed here [67,68]. Future studies could investigate whether older individuals demonstrate the same results as younger individuals.

Another methodological limitation lies within the experimental design itself and subsequent statistical design. As noted above, we chose an extreme-groups approach to this experiment (i.e., selecting the talkers that were most consistently perceived as sounding gay and straight), which allowed us to treat perceived SO as a binary variable during statistical analyses. This was done to simplify analyses given the primary focus of the current study was how listeners reacted to perceived vs actual SO given various amounts of information about the talkers' actual SO. However, the perception of SO is not as simple as a gay-to-straight binary [15,50,69]. Thus, these analyses reflect how participants were primed to respond in the experiment itself (i.e., treating perceived SO as a binary), and therefore may have limited participants' responses. Regrettably, we did not provide participants with an opportunity to describe how their own views and lived experiences may have affected their responses [50]; future experiments could provide these opportunities. Further, repeating the current experiment with talkers whose perceived SO fell on various points of perceived sexuality spectrum, rather than on the extremes, could lend further insight into the current findings.

Since this study was largely exploratory, in terms of identifying attributes associated with gay and straight men, future experiments could include additional specific attributes combined with the attributes listed by listeners [11]. To fully investigate which attributes are associated with gay and straight male talkers, future investigations could include a greater balance of positively valenced attributes and negatively valenced attributes [26], as well as more high dominance attributes and low dominance attributes [6]. The current study included more negative valence attributes (sad, mad, boring, and stuck-up) than positive valence attitudes (confident, intelligent, and outgoing) and only included one high dominance attribute. As a result, only negatively valenced attributes were associated with talkers who sounded straight (sad and old), although an equal number of negatively valenced attributes were associated with talkers who sounded gay (mad and stuck-up). Additionally, the only positively valenced attributes were associated with talkers who sounded gay (confident and outgoing), as was the only high dominance attribute (confident). Thus, we have a limited ability to draw conclusions about how listeners attribute features such as valence and dominance to talkers who sound gay and straight. Using a balanced design in future studies would rectify this shortcoming.

Finally, as mentioned previously [42], there are different attributes associated with different stereotypic groups of gay men; leather and biker gay men, for instance, are associated with tattoos and being "macho", whereas flamboyant gay men are associated with being loud and dramatic. The current study investigated gay men, in general, and presumed that certain attributes (e.g., stuck-up) would be applied to all gay men. However, future studies could investigate how listeners perceive different groups of gay men. For example, instead of just including a label of *gay* or *straight*, which was done in the current study, more specific labels could be used. It may be the case that listeners associate certain attributes with

certain types of gay men and not others. A similar experimental manipulation could also be made with straight men. We are, however, unaware of research findings that detailed which attributes are associated with different types of straight men given that most research specifically studied gay men [11,42].

## Conclusion

The current study demonstrated that perceived SO, for the most part, led to significant differences in listeners' perceptions of a variety of personal attributes between gay and straight male talkers, regardless of the talkers' actual SO. Additionally, giving listeners information about talkers' self-identified SO, whether accurate or inaccurate, significantly interacted with listeners perception of some attributes (i.e., sad and stuck-up), but not others. Together, these findings indicate that a talker's self-identified SO, whether hidden, disclosed, or falsely disclosed, has limited influence on a listener's perception of many attributes; rather, for the most part, it is the listener's perception of SO that has the greatest influence on the attributes examined in the current study. Thus, the results have potential implications for talkers perceived as gay (regardless of self-identified SO) in their day-to-day life, in relationships, and in the workplace.

## Acknowledgments

The authors wish to thank Ciara Allsup, Matthew Bradford, Dalton Davis, Jonathon Godwin, Kalie Hagedorn, Caesar Palacios, Adam Roby, and the late Jason Shuping for their help in running participants. We also wish to thank Keith Johnson, Ashley Allen, Abby Nance, Brooke Merritt, Sarah Hargus Ferguson, Shilpa Regan, Xin (Cynthia) Zhang, Gaia Gould, and Tonyetta Perry for their helpful feedback.

## Author contributions

**Conceptualization:** Erik C. Tracy, Kelly A. Charlton.

**Data curation:** Erik C. Tracy, Elizabeth D. Young.

**Formal analysis:** Erik C. Tracy, Elizabeth D. Young.

**Investigation:** Erik C. Tracy.

**Methodology:** Erik C. Tracy.

**Project administration:** Erik C. Tracy.

**Resources:** Erik C. Tracy.

**Software:** Erik C. Tracy, Elizabeth D. Young.

**Supervision:** Erik C. Tracy.

**Validation:** Erik C. Tracy, Elizabeth D. Young.

**Visualization:** Erik C. Tracy, Elizabeth D. Young.

**Writing – original draft:** Erik C. Tracy, Kelly A. Charlton.

**Writing – review & editing:** Erik C. Tracy, Elizabeth D. Young, Kelly A. Charlton.

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
