## [Decision Letter · Decision Letter 0]

24 Oct 2025

PONE-D-25-51754Judgments of American English male talkers who are perceived to sound gay or straight:  Which personal attributes are associated with each group of talkers?PLOS ONE?

Dear Dr. Tracy,

Thank you for submitting your manuscript to PLOS ONE. After careful consideration, we feel that it has merit but does not fully meet PLOS ONE’s publication criteria as it currently stands. Therefore, we invite you to submit a revised version of the manuscript that addresses the points raised during the review process.

Although the topic of the study reported in this manuscript is novel and intriguing, both reviewers raise substantial concerns needing to be addressed for this manuscript to be further considered for publication. Specifically, both R1 and R2 raise concerns about the characterization of the demographics of the participants, which may have affected their judgments in the experimental task. Moreover, R1 highlights the need for more coherent framing in the Introduction and a power analysis to justify the sample size and offers several suggestions concerning presentation of results and the focus and framing of the Discussion, and R2 points out the need for clear and consistent use of terminology. I encourage the authors to prepare a revision responsive to the reviewers' feedback, and if the authors do so, I will attempt to send the revised manuscript to the same reviewers for feedback on the extent to which their concerns are addressed.

We look forward to receiving your revised manuscript.

Kind regards,

Laura Morett

Academic Editor

PLOS ONE

Journal Requirements:

Reviewers' comments:

Reviewer's Responses to Questions

**Comments to the Author**

1. Is the manuscript technically sound, and do the data support the conclusions?

Reviewer #1: Partly

Reviewer #2: Yes

2. Has the statistical analysis been performed appropriately and rigorously?

Reviewer #1: Yes

Reviewer #2: Yes

3. Have the authors made all data underlying the findings in their manuscript fully available?

Reviewer #1: No

Reviewer #2: Yes

4. Is the manuscript presented in an intelligible fashion and written in standard English?

Reviewer #1: Yes

Reviewer #2: Yes

Reviewer #1: The manuscript titled ‘Judgments of American English male talkers who are perceived to sound gay or straight: Which personal attributes are associated with each group of talkers?’ presents a study testing how speakers’ self-identified sexual orientation, perception of sexual orientation, and ‘disclosed’ sexual orientation influence attribution of personality traits, emotions, and social categorisation (age).

I believe the research is interesting, especially the context/disclosure part that, to my knowledge, has not been tested in this way before and can expand the literature on first impressions based on actual, perceived, or disclosed sexual orientation. The topic is also of interest to scholars in different fields (e.g., linguistics, sociolinguistics, psychology, communication) due to its interdisciplinary nature. Since this is the case, the manuscript should be revised to better account for this multidisciplinarity. Below I report my concerns and suggestions, which I hope the authors will find useful for revising the manuscript and for future research.

Introduction

The introduction needs to provide a better contextualisation of this work within the current multidisciplinary literature. I appreciate that each field has looked at this topic from a different perspective, but a clearer explanation would add value to the work and show how it expands the field. When referring to the method, a clearer rationale is needed for the decision to focus on attributes. At the moment, the study is presented as an expansion of a pilot study by the authors, but this limits the contribution of the work. It is important to show how this study is embedded in the existing literature and what its contribution is, beyond expanding a single previous study.

The authors describe how the traits are relevant, but they discuss them separately and some get more attention than others in the introduction. The study focuses on a range of different attributes (i.e., sad, boring, confident, outgoing, mad, old, intelligent, and stuck-up; and subsequently anger is considered). The choice of these traits is based on a previous study (called the “pilot study”), but what is missing is an explanation of why these, and not other traits, were tested. These should be better contextualised in the multidisciplinary existing literature. The rationale for this selection is not fully explained. For instance, intelligent and confident belong to the same dimension of agency/competence and this has been tested before. So, the decision to consider them separately rather than on dimensions is needed. Moreover, one might expect the study to include traits stereotypically associated with gay and straight men, as well as neutral traits, to test whether these are differently attributed depending on actual, perceived, or disclosed sexual orientation (see McAleer et al., 2014, for how social traits were selected based on previous research; see also Madon or Blashill for work on trait attribution based on explicit sexual orientation information). Similarly, it is unclear why the attributes include these personality traits, but only a few negative emotions (sad, bored) and only one social category (age). If the goal was to test emotion and social category attribution, multiple emotions and social categories should have been included. Otherwise, the results remain quite limited, and it is unclear why sadness and age were considered more relevant than others. Finally, the authors have not considered the valence of the attributes, which can influence how they are attributed to social groups.

In relation to multidisciplinarity, attention to definitions should be improved for clarity (for instance, confidence in psychology is a trait, not an emotion). It is fine to use definitions from different fields, but this should be acknowledged somewhere in the text.

I believe the authors (who may disagree) could better introduce the study as impression formation, justify why they focused on certain attributes and not others, and clarify how these attributions are relevant for social consequences. While they do refer to social consequences, the narrative could be stronger. For instance, they could mention that intelligence and confidence may matter in educational contexts (see Taylor & Raadt on stereotyping and teaching effectiveness).

As I mentioned before, I liked the idea of testing context (or disclosure of identity). I do not believe this has been directly studied before, but there is some indirect evidence that could help build the rationale. For instance, Sulpizio et al. (2015) told participants they would hear half self-identified gay and half self-identified straight speakers, but this information did not change how speakers were categorised by voice. This suggests that first impressions may be more strongly influenced by perceived sexual orientation than by explicit information. However, Gowen and Britt (2006) found that the interplay between perceived (voice-based) and disclosed sexual orientation affected impressions and decisions, with speakers introduced as straight but sounding gay being rated more negatively. This suggests that the interaction between perceived and disclosed sexual orientation could also influence first impressions. These studies could help the authors build a stronger narrative for studying these variables together.

The section on intersectionality, when referring to age, needs further elaboration. For instance, it is not fully explained why Carnaghi et al. found that straight men were perceived as older than gay men in their work on the intersection between age and sexual orientation. This is important, as it relates to prototypes and stereotypes that are closely linked to trait attributions (see also work on voice-based intersections of sexual orientation and nationality for a similar rationale).

I would also encourage the authors to consider Sven Kachel’s work on voice-based sexual orientation categorisation and attribution of masculinity/femininity, since he considered both speaker and listener perspectives, or mention of the “straight categorisation bias,” (Lick & Johnson, 2016; Fasoli et al., 2023) which is key in distinguishing between self-identified and perceived sexual orientation. These studies could be integrated to strengthen the rationale and justify the study design and variables.

The pilot study informing the current study should be explained in more detail. This would make it clearer whether and how the results are replicated and expanded. The study should also be explicitly presented as exploratory, as no hypotheses are put forward. However, the literature on impression formation based on sexual orientation (as well as general impression formation based on voice and personality; see McAleer et al., 2014) does provide enough background to make predictions, in case the authors want to suggest some possible directions while keeping the study as exploratory.

Method

- The manuscript does not indicate whether participants’ sexual orientation was recorded. This is very important, as categorisation and perception can vary among listeners of different sexual orientations. Participants’ age and gender are also not reported. These are key demographics given the scope of the study. For instance, age should be similar across gay and straight speakers.

- Please report the percentage of correct sexual orientation categorisation for the speakers, so it is clear how perceived sexual orientation was operationalised. It would also have been good to assess perceived sexual orientation in this study to ensure consistency between participants’ categorisation and operationalisation of the variable.

- Please add a power analysis to justify the sample size and effect size for the analyses performed.

- The lack of control for regional accents is problematic, as non-standard accents trigger different first impressions (see language attitude literature) and interact with perceived sexual orientation.

- The valence of the traits should be considered to test whether it influences the attributions.

- From a psychological perspective, mad and angry are different, one is a trait and the other an emotion, this brings us back to the definitions and the different nature of the attributes considered here.

- It would be useful to clarify whether reaction times were recorded, as participants were asked to respond quickly. If so, it would be interesting to test differences in response speed depending on actual, perceived, or disclosed sexual orientation.

- I wonder whether it would be clearer to present the pairwise comparisons by keeping actual sexual orientation constant and varying the context, for instance, if a gay speaker is perceived as more stuck-up when truthfully described as gay, falsely described as straight, or when no information is provided. I think this would make it easier for the reader, though I appreciate the authors’ reporting of all comparisons.

- The section on anger attribution is unclear, as no tests of main effects or interactions are reported for anger. Without this, the results cannot be properly compared descriptively. It would also have been better to test mad and angry in the same sample for direct comparison.

- I have not found any link to a repository where the data are available. My understanding is that this is a requirement for PLOS One. If I have missed it, please ignore my comment.

Discussion

- The conclusion should elaborate more on the significant interaction between condition/context and perceived sexual orientation. At times, the text suggests that perceived sexual orientation is the only factor that matters, but for some traits, the interaction was significant.

- I would like to see more integration of how these results contribute to the multidisciplinary field of voice-based sexual orientation and impression formation.

- While I appreciated the reference to acoustic correlates in the introduction, this may be less relevant in the discussion, as no data on speakers’ acoustic features are reported.

- The explanation of the results concerning sadness needs further elaboration.

- There is a mention of studies on lesbian-sounding speakers; however, the current study focuses only on male speakers, so this seems less relevant here. The authors should elaborate on this as a limitation and on the importance of testing female speakers in future work.

- Some of the suggestions could be better supported with literature. For instance, the point about young people expecting gay men to be more outgoing based on personal experiences could be linked to the concept of familiarity tested in relation to gaydar (see Brambilla et al., 2011).

Minor points:

- Page 3, line 57: I suggest changing the sentence to clarify this is a belief: “that an individual can BELIEVE that they can perceive…”

- Across the manuscript, references should be revised, as many are not appropriate. For instance, Painter et al. (page 3, line 60) is cited when discussing attribution of different traits, but that paper focuses on stimulus length and categorisation, and only on masculinity/femininity (not tested here) not on other trait attribution. Other studies on stereotyping and impression formation would be more suitable. Similarly, on page 10, reference 15 (Mack & Munson) may not best support the statement. There are several instances of misreferencing throughout.

Reviewer #2: The manuscript is sound, legible and has data availability. Just be sure to have consistent formatting throughout the manuscript i.e. having the condition label Referenced capitalized, "SO" vs. sexual orientation, and etc.

.

Reviewer #1: No

Reviewer #2: No

---

## [Author Response · Author response to Decision Letter 1]

9 Dec 2025

We would like to thank the editor and the reviewers for their helpful suggestions and comments regarding our manuscript. The manuscript, we believe, improved with these insightful revisions. We have included most, if not all, of the indicated revisions, and the new revisions are in red font in the file “Revised Manuscript with Track Changes.” Below, we respond to each of the suggestions made by the editor and reviewers. We have also provided line numbers, so that the editor and reviewers can find these suggestions within the revised manuscript. We attempted to be as accurate as possible with the line numbers, so please excuse any mistakes.

Response to the Editor

1. We have uploaded seven files for further review: the rebuttal letter, the manuscript with track changes, the manuscript with no track changes, and the four figures. Please note that we believe that the figures should stand on their own; therefore, the figures use the term ‘sexual orientation’ instead of the SO acronym that was used in the manuscript. If the editor and/or reviewers would like us to change this, so that the figures contain the SO acronym, we are happy to oblige.

2. We have formatted the manuscript and figures in accordance with the information contained within the email that was sent to us on October 24.

Response to Reviewer #1

1. The Introduction needs to better contextualize the work within the multidisciplinary field.

The Introduction was extensively revised to meet this requirement. For instance, while we kept the discussion on perceiving sexual orientation (SO) [Lines 103-112] and attributes [Lines 113-160], we revised the section on SO and attributes [Lines 161-210]. This revised section includes a discussion on both stereotypes associated with gay and straight men [Lines 162 – 176] and discrimination in the workplace [Lines 193-210]. Additionally, we added a new section on the implications of knowing an individual’s SO [Lines 211-248]. This section includes a discussion on studies that did not reveal participants’ SO [Lines 212-221] and the emerging fields of social vision and social hearing [Lines 230-248].

2. Why were these attributes considered and not other attributes?

We further discussed the pilot study, which was used to identify the attributes in this manuscript. Briefly, we used those attributes that were identified most often by the listeners. The pilot study is discussed throughout the manuscript, and we hope that this discussion satisfies the reviewer’s question [Lines 91-92; 121-124; 323-342].

3. Why consider intelligence and confidence separately and not on a dimension?

These two attributes were both indicated by the listeners in the pilot study; the listeners differentiated between these two attributes. Here, we took a bottom-up approach and included those attributes most cited by participants in the pilot study. More information on the pilot study can be found throughout the manuscript [Lines 91-92; Lines 121-124; Lines 323-342].

4. Why not use attributes more stereotypically associated with gay and straight men?

We used the attributes that listeners indicated in the pilot study, and more information on the pilot study can be found throughout the manuscript [Lines 91-92; Lines 121-124; Lines 323-342]. We do agree with the reviewer that there is a necessity to further study these stereotypical attributes. To that end, we explained, in the Discussion, that future research studies should incorporate attributes stereotypically associated with gay and straight men [Lines 753-764].

5. Why not use neutral attributes?

As discussed previously, we used a bottom-up approach for this study by first asking listeners to identify those attributes most associated with various talkers. Unfortunately, neutral attributes were not indicated, at a high rate, by listeners; thus, we did not include them. However, in the Discussion, we do discuss further attributes that researchers could use in future studies. More information on the pilot study can be found throughout the manuscript [Lines 91-92; Lines 121-124; Lines 323-342], and more information on further attributes can be found throughout the Discussion [Lines 745-764].

6. The reviewer asked to incorporate the work of Blashill, Madon, and McAleer in the manuscript.

We found value in the work from these authors and included them within the manuscript. Blashill’s work is referenced in Line 166. Madon’s work is referenced in Lines 91, 170, 331, 747, and 764. McAleer’s work is referenced in Lines 68, 93, 129, 132, 139, and 750.

7. Why not include more social and emotion categories?

We used a bottom-up approach for this study, and we only included those attributes indicated most often by the listeners in the pilot study [Lines 91-92; Lines 121-124; Lines 323-342]. Furthermore, we did note that future studies should include more attributes, such as low valence attributes and high valence attributes [Lines 745-764].

8. Why are sadness and age more relevant than others?

Here, we were unclear what ‘others’ meant. For example, we were unclear if ‘others’ meant the other attributes investigated in this study or the other attributes (e.g., social and emotion) that the reviewer suggested? It was our intention to discuss each of the eight attributes equally. However, the rating results of the ‘sad’ attribute required more explanation, and we added this discussion in Lines 616-633.

9. The researchers haven’t considered the valence of the attributes.

Thank-you for the suggestion, and we did include a discussion on the valence of the attributes in both the Introduction [Lines 132-139; 745-752]. Additionally, we included a short discussion on another dimension: dominance [Lines 136-137; 749].

10. Terminology should be improved.

Thank-you, again, for the suggestion. We have strived to improve our terminology usage. For instance, we defined the terms attributes, attitudes, and emotions in the Introduction [Lines 114-120].

11. Possibly introduce the study as impression formation.

We agree with the reviewer and incorporated this term throughout the manuscript. For instance, it can first be found in Lines 61-69.

12. Justify why certain terms were used and why certain terms were not used.

We hoped to satisfy the reviewer’s suggestion with a greater description of the pilot study, as indicated above. Please refer to #2-5 and #7 above for a more detailed explanation.

13. Clarify why attributes are relevant to social consequences.

There is a relatively large section within the Introduction that details the social consequences of these attributes [Lines 177-210]. Here, we discussed stereotypes associated with gay and straight men, and the resulting consequences in the workplace and classroom. Additionally, there is a revised section in the Discussion that also discusses these social consequences [Lines 634-663].

14. Intelligence and confidence may matter in educational situations.

We have included Taylor and Raadt’s (2021) work on how stereotypical gay and straight male voices are perceived in an education setting [Lines 202-203; 214-216; 637-638].

15. Strengthen the rationale for testing the disclosure of the talkers’ SO.

Thank-you for this suggestion. We have included an additional section within the manuscript, ‘Knowledge of an individual’s self-identified SO’, to hopefully satisfy the reviewer’s concern [Lines 211-248]. Within this section, we discuss various auditory gaydar studies that did not disclose the talkers’ SO. Next, we discuss two studies in which knowledge of the talkers’ attributes changes how listeners perceive the talkers. Finally, we discuss both social vision and social hearing, and how these two fields provide motivation for our study.

16. The authors should consider the work on Gowen and Britt (2006) and Sulpizio et al. (2015) in terms of how the disclosure of a talker’s SO could influence the listener’s perception of them.

While we did include the work on Gowen and Britt within the paper [Lines 195-196; 647-648], we did not include the work of Sulpizio et al. (2015). However, we did include other references that we hope satisfies the reviewer’s concern. As mentioned above, we included a new section on how knowledge of an individual’s SO can affect a listener’s perception [Lines 211-248] and included within this section, we have included numerous references that hopefully strengthen our argument (e.g., Taylor & Raadt, 2021; Johnson et al., 2015).

17. The section on intersectionality needs more clarification. Why did Carnaghi et al. (2022) find that straight men were perceived as older?

We included more information about this study. For example, we included the statement that gay men were processed, by default, as young and that elderly men were processed, by default, as heterosexual [Lines 171-174].

18. Consider Sven Kachel’s work, as well as Lick and Johnson (2016) and Fasoli et al. (2023).

We did include one Kachel citation within the manuscript (Steffans, Niedlich, Kachel, & Methner, 2016) [Lines 166-169; 196-201; 660-662]. Here, we discussed Kachel’s work in terms of how a stereotype might affect one’s initial impression of another individual, and how gay and straight men might experience discrimination in the workplace. Furthermore, we included one work of Lick and Johnson within the manuscript (Johnson, Lick, & Carpinella, 2015) [Lines 232]; here we discussed their theory of social vision. Finally, we included the work of Fasoli et al. (2023) [Lines 78, 188, 198, and 216]; here we discussed how listeners place talkers in a social category and then associate various attributes with them. We also discussed how gay and straight men were perceived as equally competent and how LGBTQ+ individuals face discrimination within the workplace.

19. The pilot study needs to be explained in more detail.

We hoped to satisfy the reviewer’s comment by further discussing the pilot study. As outlined above, there is a new section within the Materials and Methods section that details the pilot study [Lines 325-344].

20. Present the study as exploratory since we are not putting forth direct hypotheses.

We agree with the reviewer in that we did not specifically predict which attributes would be associated with gay and straight talkers. Therefore, we presented the study as exploratory, and this is referenced throughout the manuscript [Lines 86, 100, 252, 574, 747].

21. Based on McAleer et al. (2014), we can offer possible directions while still keeping the study exploratory.

We included two works by McAleer: McAleer et al. (2014) [Lines 68, 132, 139, 752] and Baus et al. (2019) [Lines 93, 132,]. We discussed how listeners perceived personal attributes from talkers and the different dimensions of the social voice space.

22. Did we collect information on listener’s SO, age, and gender?

Yes, we did collect information about age and gender, but regrettably, did not collect information on SO [Lines 279-293]. We further discussed the age of our listeners both in the Methods and Materials section [Lines 279-293] and in the General Discussion [723-732], where we discussed some of the limitations of our sample. Within the manuscript, we did not report the listeners’ SO and, if memory serves correctly, there were few LGBTQ+ individuals within the listener sample. We also did not further discuss the listeners’ gender because, within the context of the manuscript, it seemed relatively less relevant, whereas age seemed more relevant and worthy of additional discussion.

23. Report the percentage of correct SO decisions, so that we know how SO was operationalized.

We discussed how the talkers were selected for the experiment [Lines 306-317]. We discussed the results of Tracy et al. (2015) in which each talker was rated by how confident listeners were of the talker’s SO. Next, we selected those talkers who fell at the extremes for inclusion in the present experiment. By extremes, we mean those self-identified gay talkers whom listeners labeled gay most often, those self-identified gay talkers whom listeners labeled straight most often, those self-identified straight talkers whom listeners labeled gay most often, and those self-identified straight talkers whom listeners labeled straight most often.

24. Please add a power analysis to justify the sample size and effect size for the analyses performed. Please elaborate on the effect sizes.

A sample size calculation for a poisson regression was completed using G*Power 3.1. Based on a predicted medium effect size, a beta of 0.8, and alpha of 0.05, the sample size of listeners needed for this experiment was calculated to be 164. However, the use of a nested data structure (i.e., stimuli nested within talkers nested within listeners) increases the standard error of a dataset. We estimated the interclass correlation between talkers and listeners to be approximately 0.1, which introduces approximately an 8-10% margin of error for the proposed sample of 24 talkers (Gelman & Hill, 2007); thus an additional ~15 participants were needed to account for this error. Our target sample size was thus ~180.

The recruitment method for this study was to offer course credit to undergraduate classes with a size ranging anywhere from 20-40 students. Based on the power analysis, we determined to offer this experiment to students in 9 classes (9 x 20 = 180) to ensure adequate power. Given that all students in each class were given the option to participate in the experiment, we expected this sampling method to result in our minimum recruitment target (i.e., 180), but anticipated that it may give us participants beyond this number.

The rate ratios (RR) provided throughout the manuscript were provided as measures of effect size as demonstrate the magnitude of the effect in a way that is directly comparable between effects and studies. Please let us know if you feel another effect size would be more appropriate.

Citation:

Gelman, A. & Hill, J. (2007). Data Analysis Using Regression and Multilevel/Hierarchical Models. Cambridge University Press.

25. Lack of control for regional accents is problematic as non-standard accents can affect first impressions.

We addressed the reviewer’s concern within the Materials and Methods section [Lines 295-305]. Talkers were required to be from the state of Ohio, but we did not control for regional accents within the state. However, the first author, who is a trained phonetician, did not notice any differences in regional accents of the talkers and, presumably, the listeners would not notice this either. Moreover, it was difficult to recruit gay talkers for the original study (e.g., Tracy et al., 2015), and those authors did not have the luxury of only sampling gay talkers who grew up in the Columbus, Ohio metropolitan area; therefore, the authors needed to expand their geographic region to the entire state of Ohio to recruit a large enough sample of gay talkers. We hope that these explanations satisfy the reviewer’s concerns.

26. Valence of the attributes should be considered as to whether it influences the attributions.

Within the Introduction, we included a more detailed section on the dimensions of attributes [Lines 132-139] and discussed where our chosen attributes fall within these dimensions. We admit that our attributes were not balanced along these dimensions. For example, we did not include equal numbers of high valence attributes and low valence attributes. We also did not include equal numbers of high dominance attributes and low dominance attributes. This is a shortcoming in the current exploratory study, and we discussed including additional attributes in future experiments within the General Discussion [Lines 747-754].

27. Mad and angry are different. Bring us back to definitions and different natures of attributes.

We took care to use consistent and clear definitions of both emotions and attitudes [Lines 114-120]. Furthermore, we also noted that the listeners in the pilot study described the talkers as mad (i.e., they did not use the term angry), and we wanted to utilize the terminology used by the listeners [Lines 126-

---

## [Decision Letter · Decision Letter 1]

17 Feb 2026

PONE-D-25-51754R1Judgments of American English male talkers who are perceived to sound gay or straight:  Which personal attributes are associated with each group of talkers?PLOS One?

Dear Dr. Tracy,

We look forward to receiving your revised manuscript.

Kind regards,

Laura Morett

Academic Editor

PLOS One

Journal Requirements:

Reviewers' comments:

Reviewer's Responses to Questions

**Comments to the Author**

Reviewer #2: All comments have been addressed

Reviewer #3: All comments have been addressed

2. Is the manuscript technically sound, and do the data support the conclusions?

Reviewer #2: Yes

Reviewer #3: Yes

3. Has the statistical analysis been performed appropriately and rigorously?

Reviewer #2: Yes

Reviewer #3: Yes

4. Have the authors made all data underlying the findings in their manuscript fully available?

Reviewer #2: Yes

Reviewer #3: No

5. Is the manuscript presented in an intelligible fashion and written in standard English?

Reviewer #2: Yes

Reviewer #3: Yes

Reviewer #2: Thank you for the revised manuscript. This revision further contextualizes the study with relevant theory and implications (e.g., social hearing, workplace/classroom social consequences) and has substantially improved from the prior draft. I reviewed the authors’ response letter alongside the revised manuscript and found my major concerns to be addressed. I also reviewed the authors’ response to Reviewer 1’s many insightful feedback points. The authors report substantive revisions that address Reviewer 1’s major concerns (e.g., stronger conceptual framing, expanded pilot study description/rationale, and added reporting items like the repository link and power analysis). I did not see anything in Reviewer 1’s list that would, on its face, change my recommendation, though I defer to the editor on any remaining points requiring additional verification. The remaining issues are copyedits/consistency only and do not affect the interpretation of the results. Thus, I believe the manuscript is suitable for publication pending minor editorial/copyediting revisions.

Comment R2-1, 5, 6, 7

The manuscript is in noticeably better shape after the terminology updates and grammatical clean-up. I appreciate the thoroughness of these revisions and am satisfied with the changes. I did catch a few remaining consistency items that are easy to fix (e.g., standardizing “outgoing” vs. “out-going,” and ensuring the manuscript consistently uses “boring” rather than “bored” as the label).

Comment R2-2

I appreciate the authors’ effort to engage the literature more directly here. The added context is reassuring with respect to the study’s design, and I agree with the authors’ decision to prioritize brevity given the aims of the paper. I am satisfied with the current state of this section.

Comment R2-3

I appreciate the inclusion of Niedzielski (1999) and the broader contextualization around listener perception and labeling. While I would have liked a slightly more explicit tie back to how this literature informs the present study’s predictions and interpretation, the addition strengthens the conceptual grounding, and I am satisfied with it in its current form.

Comment R2-4

I am appreciative of the additional detail provided here. The discussion of Tracy & Johnson (2015) (i.e., similar judgments across sites) and the inclusion of campus demographic context serve as a reasonable proxy for addressing the concern. I am satisfied that appropriate precautions and contextual framing have been considered.

[48] bored -> boring (consistency)

[106] Add “as” in between “such” and “pitch”

[178] Not sure what [eo] means here, assuming it’s a typo

[201] Delete “a” between “seeing” and “photographs”

[289] participants -> participants’

[515] False Referenced -> Falsely Referenced

[536] False Referenced -> Falsely Referenced

[603-604] transpose “was” and “SO”

[621] of -> or

[631] word -> words (?, I think the plural was meant here)

[773] talkers’ -> talker’s

Reviewer #3: This is a highly interesting study in which the authors explored whether listeners associated additional personal attributes with these types of talkers and whether different contexts (e.g., listeners being informed of the talker’s sexual orientation) influenced the strength of such attribute associations made by listeners. The findings revealed that sexual orientation per se did not affect attribute perception, yet informing listeners of the talker’s sexual orientation (whether the information was accurate or not) exerted an impact on their perceptions. This finding is of great significance for an in-depth understanding of the mechanisms underlying the formation of listener impressions. The authors have responded positively to the comments raised by the reviewers in the first round of review and made corresponding revisions accordingly. In my opinion, this paper has basically met the publication requirements and is acceptable for publication upon completion of minor revisions.

Minor Revision Requirements

1. The manuscript requires a thorough proofread, as several errors may impede readers’ comprehension. For instance, Line 289: The participants mean age was 19.6 (grammatical error); Line 351: 24 talkers X 2 utterances X 4 attributes (using "X" as a multiplication sign, which is non-standard in academic writing).

2. The participants recruited for the experiment are relatively young (with a mean age of 19.6 years). Perceptual differences may exist among participants of different age groups and with varying social experience, and this limitation should be explicitly addressed in the Limitations section.

3. Ensure terminology consistency across all tables and figures. The authors noted in their response to reviewers that they have addressed consistency issues (e.g., using "stuck-up" instead of "stuck up"). A final careful proofread of all tables and figures is therefore recommended, as consistent terminology in the final version will enhance the paper’s professionalism.

4. Verify the accessibility of the materials available at the link: https://osf.io/bc2wm/overview?view_only=8da8397267a04e21a34f1c367f626a9b. Currently, the page loads as a blank screen.

5. The subheadings of the paper can be further refined to improve readability. For example, the subheading Results: Aim 1 could be revised to include a concise description of Aim 1 as the heading.

6. Line 428: seen in Table 2 and Fig 1 – "Fig" should be spelled out in full as "Figure".

7. The paper’s structure lacks clarity and appears disorganized. For example, Experiment 1a includes a combined Results and Discussion section, while Experiment 1 only has a standalone Results section; additionally, the sequencing of Experiment 1 followed by Experiment 1a is confusing and requires clarification.

8. Lines 747–754: Although the study did not balance high- and low-valence attributes, the authors may still provide a relevant discussion of high- and low-valence attributes here.

9. The expanded discussion on the "sad" attribute, as requested by the reviewers, is valuable. The explanation that different attributes may activate distinct pathways in the social hearing model is particularly insightful. To further improve readability, the conclusion of this paragraph should be more succinct, concisely summarizing the key finding as to why the "sad" attribute exhibited different patterns from other attributes; this will create a clearer logical bridge to the subsequent discussion points.

.

Reviewer #2: No

Reviewer #3: No

---

## [Author Response · Author response to Decision Letter 2]

3 Mar 2026

We would like to thank everyone for their insightful comments that helped us to improve our manuscript. The new revisions are in red font in the file “Revised Manuscript with Track Changes.” Below, we’ve responded to each of the suggestions made by the reviewers. Additionally, we’ve attempted to provide line numbers, so that the editor and reviewers can easily find the changes. We apologize if there are any discrepancies with the line numbers.

Response to Reviewer #1

1. Please be consistent with out-going versus outgoing.

We have changed all instances of “out-going” to “outgoing”; this is consistent throughout the manuscript.

2. Please be consistent with boring versus bored.

We have changed all instances of “bored” to “boring”; this is consistent throughout the manuscript.

3. The reviewer provided several small grammatical changes to make, and we have fixed all these small grammatical changes, as well as any other typos or grammatical changes we noticed during our read-through. Specifically, the reviewer noted they thought that [eo] was a typo. After checking this, we discovered that it was not a typo, but rather a missing reference. We have changed this to the correct reference [Line 178].

Response to Reviewer #2

1. Line 289, The participants mean age was 19.6 and this was not grammatical.

We assumed that the reviewer meant that we should include an apostrophe after "participants”, and we included this.

2. Line 351: 24 talkers X 2 utterances X 4 attributes (using "X" as a multiplication sign, which is non-standard in academic writing)

We deleted these sections (e.g., 24 talkers X 2 utterances X 4 attributes) in both Experiment 1 and Experiment 1a. Instead, both sections state the number of stimuli, trials, or lists [Lines 324, 332, 351, 361, 555].

3. The participants recruited for the experiment are relatively young (with a mean age of 19.6 years). Perceptual differences may exist among participants of different age groups and with varying social experience, and this limitation should be explicitly addressed in the Limitations section.

We agree that perceptual differences between age groups likely exist; we addressed differences in the social perception of LGBTQ+ communities in lines 708 – 728, as well as age-related differences in emotion perception in lines 729 – 738. In response to this comment, we further clarified that our listener sample is relatively young [Lines 707-708], and added information regarding other factors that may affect perceptual differences between age groups [Lines 736-738]

4. The data has not been made publicly available / Page is blank.

We apologize that the link was not working for the reviewer; it is possible that the link may have erroneously included an extra period at the end. All three authors independently verified that the supplied link is now working correctly. If you click on the link, it will take you to a page with two folders – Data and Statistical Analyses. Files are included in each of the two folders, even though the folders are indicated as 0 B. If you click on the Data folder, there are two publicly available data sets. If you click on the Statistical Analyses folder, there are four publicly available data files that contain all the statistical analyses for the current project.

5. The subheadings of the paper can be further refined to improve readability; Results Aim 1 changed to a concise description of Aim 1.

We changed the subheadings to align with the reviewer’s suggestion [Lines 419 – 420; 469].

6. Line 428: seen in Table 2 and Fig 1 – "Fig" should be spelled out in full as "Figure".

We changed all instances of the word “Fig” to the word to “Figure.”

7. The paper’s structure lacks clarity and appears disorganized; Experiment 1a includes a combined Results and Discussion section, while Experiment 1 only has a standalone Results section.

We deleted the “Results and Discussion” from Experiment 1a. It now reads “Results.” [Line 560]. As a result, the subheadings in Experiment 1 and Experiment 1a align more clearly with one another.

8. The sequencing of Experiment 1 and Experiment 1a is confusing and needs clarification.

To address this, we clarified earlier in the manuscript that anger and mad will be explored in Experiment 1a [Line 129]. Furthermore, we clarified the objectives of both Experiment 1 and Experiment 1a [Lines 252, 262, 270-271].

9. Lines 747–754: Although the study did not balance high- and low-valence attributes, the authors may still provide a relevant discussion of high- and low-valence attributes here.

We now include a discussion of the distribution of positive/negative valence in the attributes that were associated with gay and straight talkers and expand upon how a balanced design in future studies could further elucidate how listeners use features such as valence and dominance when making judgements of listener attributes [Lines 761-768],

10. Discussion on Sad Attribute; Conclusion of paragraph needs to be succinct; Summarize the key findings as to why sad demonstrated a different pattern.

We summarized the discussion on “sad” by highlighting the simplest, and perhaps, most likely explanation [Lines 634-637].

---

## [Editor Report · Decision Letter 2]

26 Mar 2026

Judgments of American English male talkers who are perceived to sound gay or straight:  Which personal attributes are associated with each group of talkers?

PONE-D-25-51754R2

Dear Dr. Tracy,

We’re pleased to inform you that your manuscript has been judged scientifically suitable for publication and will be formally accepted for publication once it meets all outstanding technical requirements.

Kind regards,

Laura Morett

Academic Editor

PLOS One

Additional Editor Comments (optional):

I thank the authors for addressing the remaining minor comments raised by the reviewers. I am now pleased to recommend acceptance of the manuscript for publication in PLOS One.
---

## [Editor Report · Acceptance letter]

PONE-D-25-51754R2

PLOS One

Dear Dr. Tracy,

I'm pleased to inform you that your manuscript has been deemed suitable for publication in PLOS One. Congratulations! Your manuscript is now being handed over to our production team.

Kind regards,

on behalf of

Dr. Laura Morett

Academic Editor

PLOS One